# Atomic high-spin cobalt(II) center for highly selective electrochemical CO reduction to CH₃OH

Jie Ding [1,2,9], Zhiming Wei[1,9], Fuhua Li[2,9], Jincheng Zhang[2], Qiao Zhang[1], Jing Zhou [3] ✉, Weijue Wang[4], Yuhang Liu[5], Zhen Zhang[6], Xiaozhi Su [7], Runze Yang[6], Wei Liu [4], Chenliang Su [8] ✉, Hong Bin Yang[5] ✉, Yanqiang Huang [4], Yueming Zhai [1] ✉ & Bin Liu [2] ✉

In this work, via engineering the conformation of cobalt active center in cobalt phthalocyanine molecular catalyst, the catalytic efficiency of electrochemical carbon monoxide reduction to methanol can be dramatically tuned. Based on a collection of experimental investigations and density functional theory calculations, it reveals that the electron rearrangement of the Co 3d orbitals of cobalt phthalocyanine from the low-spin state (S = 1/2) to the high-spin state (S = 3/2), induced by molecular conformation change, is responsible for the greatly enhanced CO reduction reaction performance. *Operando* attenuated total reflectance surface-enhanced infrared absorption spectroscopy measurements disclose accelerated hydrogenation of CORR intermediates, and kinetic isotope effect validates expedited proton-feeding rate over cobalt phthalocyanine with high-spin state. Further natural population analysis and density functional theory calculations demonstrate that the high spin Co²⁺ can enhance the electron backdonation via the $d_{xz}/d_{yz}$−2π* bond and weaken the C-O bonding in *CO, promoting hydrogenation of CORR intermediates.

Electrochemical CO₂ reduction reaction (CO₂RR), as a promising strategy for CO₂ mitigation and transformation, has been extensively studied over the past few decades[1–5]. Liquid products, especially CH₃OH, possess significant advantages because of their high energy density and ease of storage[6,7]. However, few catalysts are capable to convert CO₂ to products other than CO or HCOOH with high activity and selectivity. Electrochemical CO reduction reaction (CORR) emerges as an encouraging approach to produce other products[8–12]. Among the reported electrocatalysts, Cu is the mostly used one for CORR, but its selectivity towards a single product is poor because of the complicated chemical states of Cu surface[13–15]. Till now, developing efficient electrocatalysts for highly active and selective CORR remains challenging yet urgent.

Single atom catalysts (SACs) could act as a promising platform to selectively catalyze CORR due to the absence of consecutive active sites[16–18]. However, few SACs have been reported to reduce CO to

[1]The Institute for Advanced Studies, Wuhan University, Wuhan 430072, China. [2]Department of Materials Science and Engineering, City University of Hong Kong, Hong Kong SAR 999077, China. [3]Shanghai Institute of Applied Physics, Chinese Academy of Sciences, Shanghai 201800, China. [4]CAS Key Laboratory of Science and Technology on Applied Catalysis, Dalian Institute of Chemical Physics, Chinese Academy of Sciences, Dalian 116023, China. [5]School of Materials Science and Engineering, Suzhou University of Science and Technology, Suzhou 215009, China. [6]China Astronaut Research and Training Center, Beijing 100094, China. [7]Shanghai Synchrotron Radiation Facility, Shanghai Advanced Research Institute, Chinese Academy of Sciences, Shanghai 201204, China. [8]International Collaborative Laboratory of 2D Materials for Optoelectronics Science and Technology of Ministry of Education, Engineering Technology Research Center for 2D Materials Information Functional Devices and Systems of Guangdong Province, Institute of Microscale Optoelectronics, Shenzhen University, Shenzhen 518060, China. [9]These authors contributed equally: Jie Ding, Zhiming Wei, Fuhua Li. ✉e-mail: zhoujing@sinap.ac.cn; chmsuc@szu.edu.cn; yanghb@usts.edu.cn; yueming@whu.edu.cn; bliu48@cityu.edu.hk

generate liquid products because SACs usually show weak adsorption towards *CO, which is detrimental to the subsequent hydrogenation reaction. Moreover, the real structure of experimentally synthesized SACs is rather complex that contains a large variety of single-atom centers with different coordination environments, which greatly complicates the understanding of the structure-performance relationship. On the other hand, molecular catalysts with precisely defined coordination structures offer a model system to probe the reaction mechanism and reveal the catalyst structure-performance relationship[19–23]. Cobalt phthalocyanine (CoPc) molecular catalyst showed the capability to electrochemically reducing CO to $CH_3OH$, however, the current density and the $CH_3OH$ selectivity are still unsatisfactory[24]. The spatial structure of binuclear cobalt phthalocyanine is very different from that of mononuclear cobalt phthalocyanine, which is anticipated to result in distinct crystal field and thus different catalytic performance, making binuclear cobalt phthalocyanine an interesting candidate to be investigated in CORR.

In this work, two model catalysts were constructed by anchoring cobalt phthalocyanine and binuclear cobalt phthalocyanine (M-CoPc and B-CoPc) on nitrogen-doped carbon support. It is found that the B-CoPc can be transferred from low-spin state (LS, 1/2) to high-spin state (HS, 3/2) after thermal treatment, and the HS B-CoPc is able to catalyze CORR to methanol much more effectively than LS M-CoPc and B-CoPc. The $CH_3OH$ partial current density reaches about 35 mA/cm² at −0.8 V (vs. RHE) with a methanol Faradaic efficiency (FE) of 50%. *Operando* attenuated total reflectance surface-enhanced infrared absorption spectroscopy (ATR-SEIRAS) measurements and density functional theory (DFT) calculations disclose that the high spin $Co^{2+}$ in HS B-CoPc can enable electron accumulation on *CO that will greatly weaken the C−O bonding, promoting the hydrogenation of CORR intermediates ($*CO/*CH_xO$), which lead to boosted CORR activity and selectivity.

## Results

### Structure characterization

CoPc (M-CoPc) and binuclear CoPc (B-CoPc) were first anchored onto nitrogen-doped carbon (NC) via π-π interaction to obtain the model SACs (M-CoPc-RT and B-CoPc-RT), which were then thermally treated at 400 °C in an Ar atmosphere to enhance the structure stability of the composites (named as M-CoPc-400 and B-CoPc-400, respectively, Fig. 1a). Figure S1a, b and Figure S2a, b show mass spectrometry (MS) and nuclear magnetic resonance (NMR) spectra of M-CoPc and B-CoPc. Both M-CoPc-400 and B-CoPc-400 display three-dimensional, ultrathin sheet-like structures with all elements uniformly distributed as shown in the scanning electron microscope (SEM), transmission electron microscope (TEM) images, and energy-dispersive X-ray spectroscopy (EDX) elemental mappings (Figs. S1, S2). Aberration-corrected high-angle annular dark-field scanning transmission electron

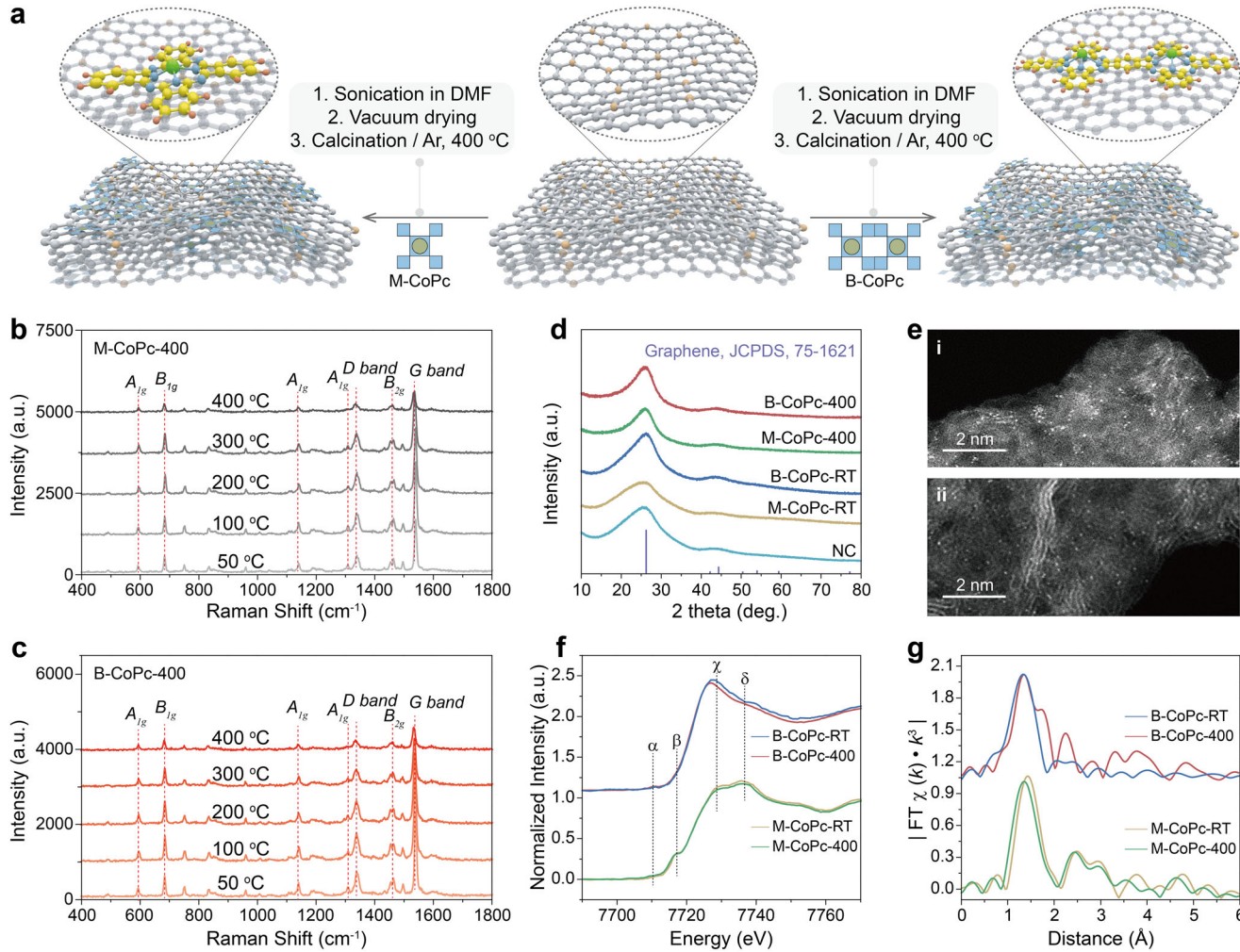

**Fig. 1 | Structural characterization. a** Schematic of the synthesis process for M-CoPc-RT/400 and B-CoPc-RT/400. In situ Raman spectra of **b** M-CoPc-RT and **c** B-CoPc-RT recorded during the thermal treatment process. **d** X-ray diffraction patterns. **e** The high magnification HAADF-STEM image of (i) M-CoPc-400 and (ii) B-CoPc-400 (scale bar, 2 nm). **f** Co K-edge XANES spectra before and after annealing. **g** The corresponding Fourier transformation (FT)-EXAFS spectra.

microscopy (HAADF-STEM) was further used to directly observe the atomically dispersed Co atoms, which appear as bright spots as shown in Fig. 1e. The structure evolution of M-CoPc-RT and B-CoPc-RT during thermal treatment was investigated by in situ Raman spectroscopy. As shown in Fig. 1b, c, the Raman spectra of M-CoPc-RT and B-CoPc-RT exhibit identical characteristic features[25]. X-ray diffraction (XRD) reveals similar structures of M-CoPc and B-CoPc with only carbon diffraction peaks (Fig. 1d). Moreover, all characteristics of CoPc were preserved during the thermal treatment, indicating that the structure of CoPc did not change significantly before and after the thermal treatment.

The valence states of Co in M-CoPc-400 and B-CoPc-400 were examined by X-ray photoelectron spectroscopy (XPS). Figure S3 shows the C 1s, N 1s, and Co 2p XPS spectra. The Co 2p XPS peaks for M-CoPc-RT and B-CoPc-RT are obviously different in peak shape and binding energy (Fig. S3c), indicating different interactions between M-CoPc-RT and B-CoPc-RT with NC support. In addition, the binding energy of Co 2p for M-CoPc-400 and B-CoPc-400 are similar to that for M-CoPc-RT and B-CoPc-RT, matching well with the in situ Raman spectroscopy results, suggesting minimal structural change of CoPc during thermal treatment.

X-ray absorption near edge structure (XANES) and extended X-ray absorption fine structure (EXAFS) characterizations were employed to further explore the chemical states and local coordination structure of the Co atom. Figure 1f shows the Co K-edge XANES spectra of M-CoPc-RT/400 and B-CoPc-RT/400. The Co K-edge XANES spectrum of M-CoPc-RT/400 is similar to that of CoPc, indicating that the M-CoPc-RT/400 possesses the same Co oxidation state and D4h symmetry as CoPc[26]. However, the intensity of peak B ($1s{\rightarrow}4p_z$ transition) which is a fingerprint of square-planar metal-N4 structure is slightly weaker in B-CoPc-RT/400, suggesting a distorted D4h symmetry of the Co atom[27]. Furthermore, the B-Co-RT/400 displays a remarkably larger intensity ratio of peak C to peak D

($I_C/I_D$) than M-CoPc-400 (peak C and peak D can be assigned to the $1s{\rightarrow}4p_{x,y}$ transitions and multiple scattering processes, respectively). Increased $I_C/I_D$ had been shown to be beneficial towards promoting electrochemical reactions[28,29]. In addition, C and N K-edge XANES spectra also indicate that M-CoPc-400 and B-CoPc-400 compose of the similar species (Fig. S4), in agreement with the XPS results. Figure 1g displays the Fourier-transformed EXAFS spectra. The peak at 1.55 Å for M-CoPc-RT/400 results from the scattering of the first Co-N shell. Notably, the first shell Co-N distance of B-CoPc-400 is slightly longer than that of B-CoPc-RT, which may be caused by molecular distortion induced by heat treatment[30]. Table S1 summarizes the EXAFS fitting results (coordination number and length of Co-N bond, Fig. S5).

## Electrochemical CORR performance

CORR over M-CoPc-RT/400 and B-CoPc-RT/400 were assessed by cyclic voltammetry (CV) in an Ar-saturated or CO-saturated 0.5 M KOH solution. As shown in Figure S6c, d, CV curves of M-CoPc-400 and B-CoPc-400 display the clear redox peak at about −0.51 V (vs. RHE) in Ar-saturated 0.5 M KOH solution, corresponding to Co(II) reduction to Co(I)[31]. Compared to that in Ar-saturated 0.5 M KOH solution, the onset potential of the redox peak in CO-saturated 0.5 M KOH solution clearly shifted to lower cathodic potentials, suggesting that CO activation took place on the Co(II) center of M-CoPc-400 and B-CoPc-400, instead of Co(I).

Besides, the two thermally treated catalysts display much higher current densities than the untreated ones. The cathodic redox peak current of M-CoPc-400 and B-CoPc-400 exhibit a linear dependence on CV scan rate (Fig. S7a, b), suggesting that the CORR on M-CoPc-400 and B-CoPc-400 is a surface reaction-controlled process. In addition, the linear sweep voltammetry (LSV) curves (Fig. 2b) further show that the B-CoPc-400 exhibits a much better CORR performance than M-CoPc-400. Figure 2c compares the potential-dependent product

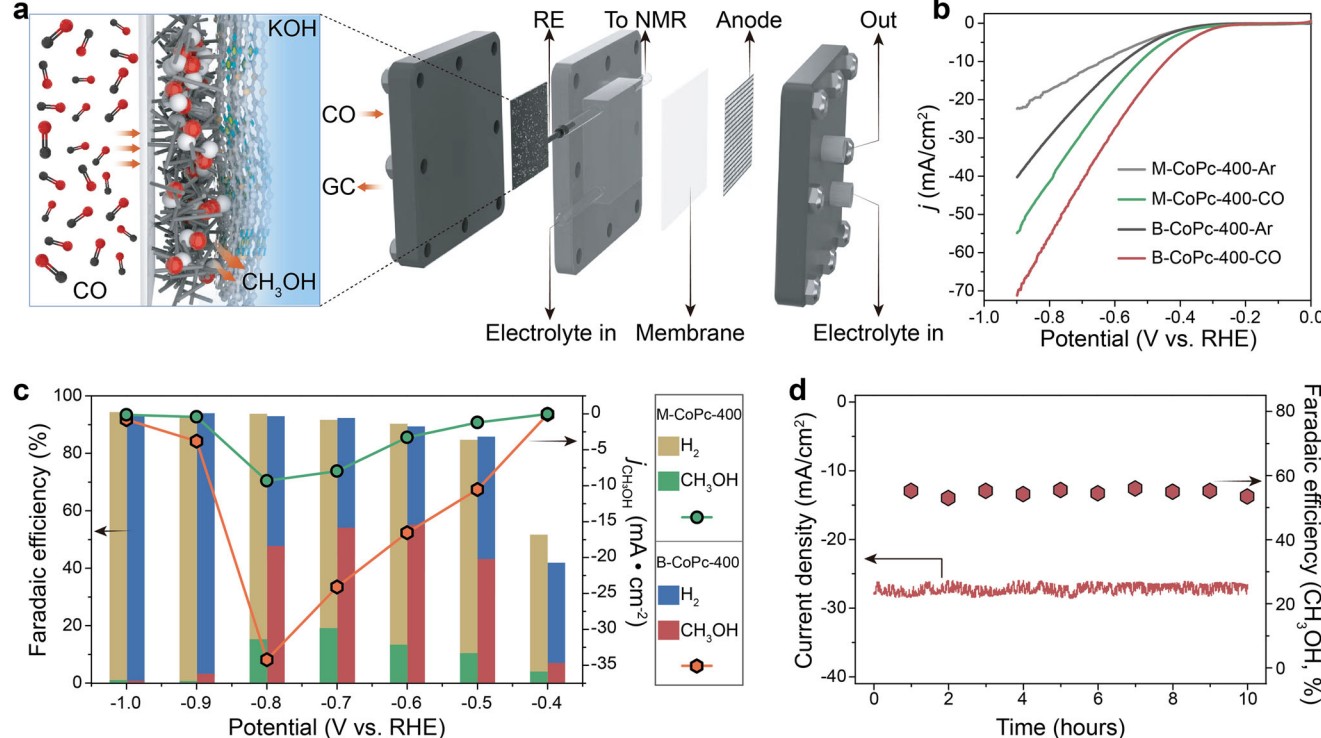

**Fig. 2 | Catalytic performance. a** Schematic diagram showing the MEA for CORR. **b** LSV curves acquired in Ar and CO-saturated 0.5 M KOH solution on a carbon paper at a scan rate of 5 mV/s. **c** Potential-dependent product selectivity for CORR catalyzed by M-CoPc-400 and B-CoPc-400. **d** Stability of M-CoPc-400 recorded at −0.7 V (vs. RHE). The measurement was performed at the condition of 1 atm CO and room temperature in CO-saturated 0.5 M KOH.

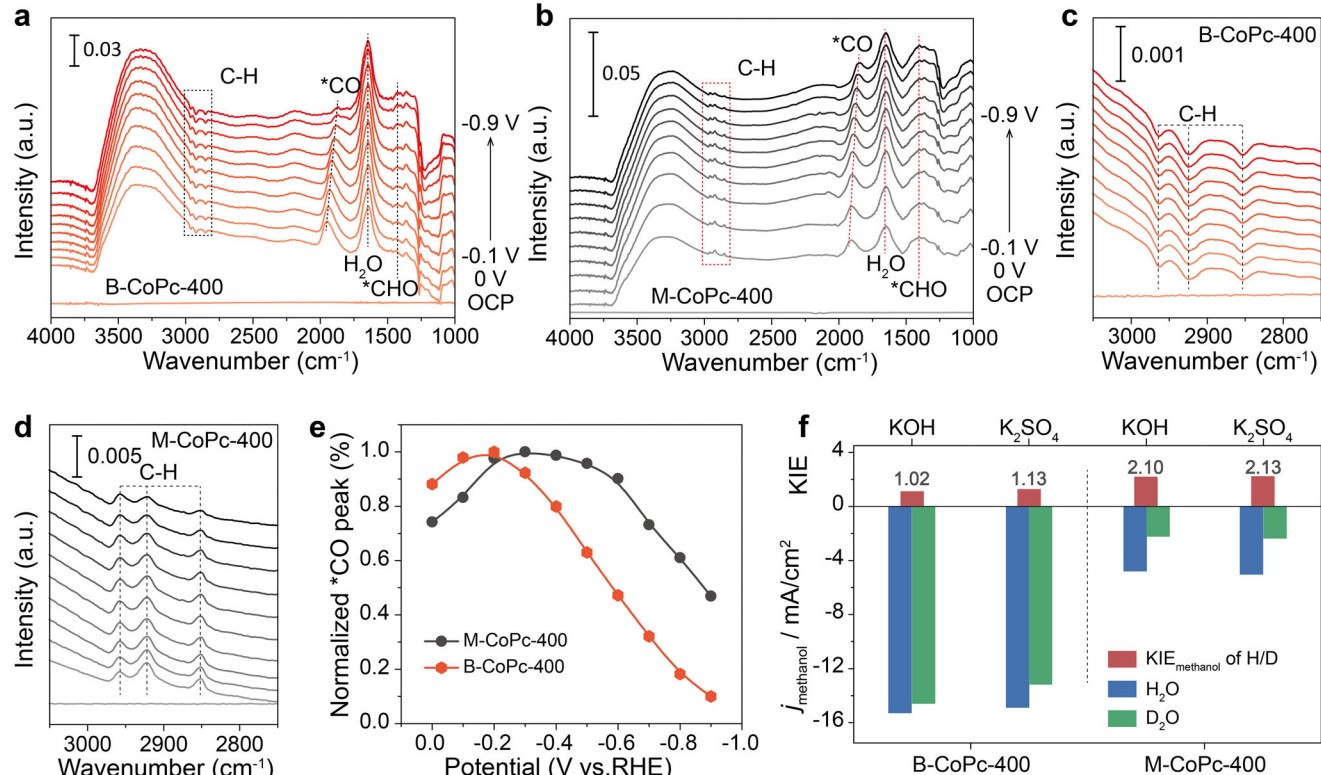

**Fig. 3 | *Operando* ATR-SEIRAS and KIE measurements.** *Operando* ATR-SEIRAS spectra of CO reduction over **a**, **c** B-CoPc-400 and **b**, **d** M-CoPc-400 in CO-saturated 0.5 M KOH. The spectra were collected at constant potentials with 0.1 V interval in the cathodic direction from OCP to −0.9 V (vs. RHE). The *operando* ATR-SEIRAS spectra recorded in Ar-saturated 0.5 M KOH are shown in Fig. S13. **e** Potential dependent CO stretching peak intensity. **f** KIE of H/D in CORR to $CH_3OH$ at −0.6 V (vs. RHE) over B-CoPc-400 and M-CoPc-400.

distribution of CORR over M-CoPc-400 and B-CoPc-400. The main product on M-CoPc-400 is $H_2$, with a Faradaic efficiency (FE) over 80% in the potential range from −0.5 V to −1.0 V (vs. RHE). The maximum FE for $CH_3OH$ over M-CoPc-400 is ~19% at −0.7 V (vs. RHE), matching well with the early work[21].

On the contrary, $CH_3OH$ becomes the dominate product over B-CoPc-400 in the potential range of −0.5 V to −0.8 V (vs. RHE), illustrating that the B-CoPc-400 could catalyze CORR much more effectively. [1]H 1D NMR spectra of products and DMSO at different potentials (vs. RHE) for B-CoPc-400 and M-CoPc-400 are shown in Fig. S8a, b. The $CH_3OH$ partial current density can reach about 35 mA/cm² at −0.8 V (vs. RHE) with a FE of 50%, and the maximum $CH_3OH$ FE of 57% appears at −0.6 V (vs. RHE) over B-CoPc-400. The yields of $CH_3OH$ produced over B-CoPc-400 using isotopically labeled [13]CO and [12]CO at multiple points for different potentials (−0.5 V, −0.6 V, −0.7 V vs. RHE) are consistent quantitatively (Fig. S9). Table S2 displays FE of all the products during CORR. The calculated $CH_3OH$ turnover frequency (TOF) over B-CoPc-400 and M-CoPc-400 are displayed in Fig. S10, suggesting that the loading amount of Co plays a negligible role in affecting the electrochemical performance. Table S3 compares the CORR performance between our prepared catalyst and those reported in the literature.

In addition, the influence of treatment temperature and types of support on catalytic performance were also investigated, and the results are summarized in Fig. S11, showing that the treatment temperature has a significant effect on the catalytic performance, while the influence of support is insignificant. Moreover, the B-CoPc-400 could produce $CH_3OH$ with a FE of 53% at −0.7 V (vs. RHE) with negligible current loss for 10 h, corroborating its excellent catalytic durability (Fig. 2d). Figure S12 shows the HRTEM, HAADF-STEM and XAS results

after CORR stability test, corroborating the robust stability of B-CoPc-400.

## Mechanistic insights into proton-feeding

To understand the different CORR catalytic activity as well as the CORR mechanism over M-CoPc-400 and B-CoPc-400, *operando* ATR-SEIRAS was conducted to probe the reaction intermediates. Figure 3a, b shows the *operando* ATR-SEIRAS spectra of CORR over B-CoPc-400 and M-CoPc-400. Three peaks at ~1870 cm⁻¹, ~1650 cm⁻¹, and ~1400 cm⁻¹ are clearly identified, which can be assigned to *CO, $H_2O$, and *CHO, respectively[32–34]. Notably, both M-CoPc-400 and B-CoPc-400 could adsorb CO at 0 V (vs. RHE). The *CO vibrational frequency on M-CoPc-400 is slightly lower than that on B-CoPc-400 (1911 and 1951 cm⁻¹ at 0 V vs. RHE), indicating a slightly lower binding strength of B-CoPc-400 towards CO. With increasing cathodic potential from 0 to −0.9 V (vs. RHE) over M-CoPc-400 and B-CoPc-400, *CO vibrational frequency red-shifted with a decrease in peak intensity due to the Stark effect, and the intensity of the *CO stretching peak over B-CoPc-400 decreased faster with increasing cathodic potential as compared to that over M-CoPc-400 (Fig. 3e), indicating faster consumption of *CO over B-CoPc-400. When $D_2O$ was used instead of $H_2O$, the position of CHO moved by 25 cm⁻¹ towards smaller wavenumbers due to isotopic redshift caused by the mass effect, confirming that the peak at ~1400 cm⁻¹ can be ascribed to *CHO (Fig. S14). Compared to M-CoPc-400, the peak intensity of *CHO is always lower on B-CoPc-400 in the entire potential range (from 0 to −0.9 V vs. RHE), indicating lower coverage of *CHO species on the B-CoPc-400 surface, which may result from the fast hydrogenation of *CHO to the final product. It is worth to note that the C−H bands between 3000 and 2800 cm⁻¹ originating from different vibrational modes in −$CH_3$ and $CH_2$ over B-CoPc-400 are

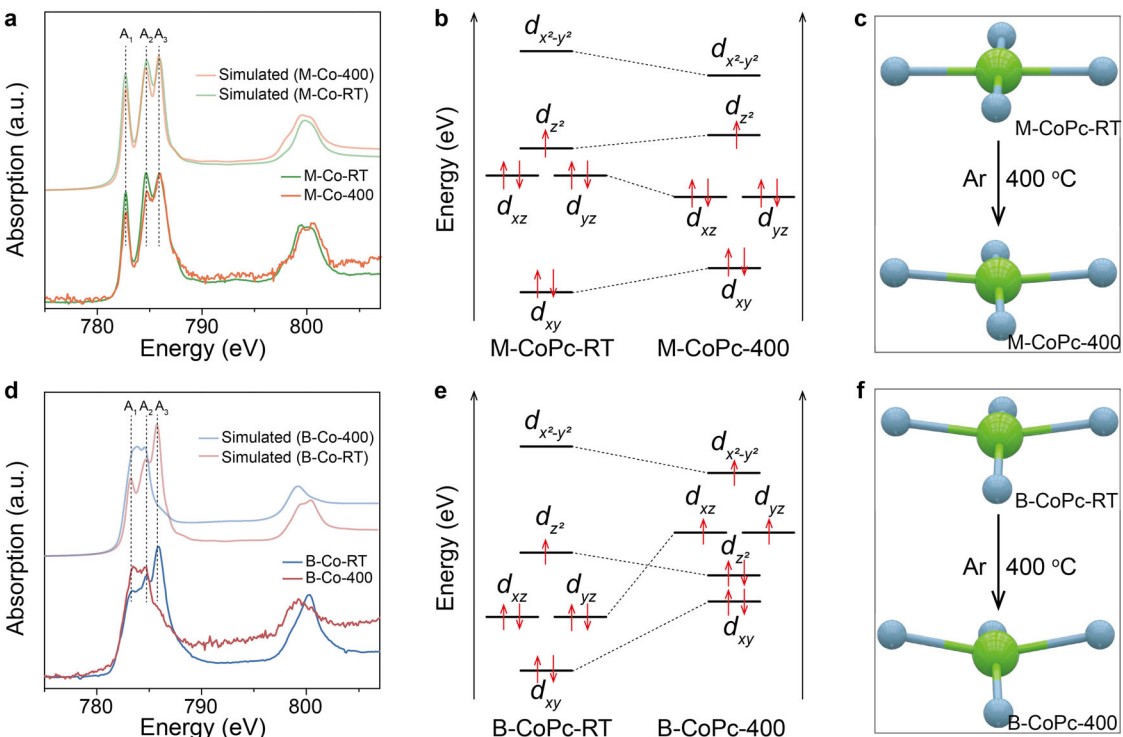

**Fig. 4 | Analysis of electronic structure. a, d** The experimental and simulated cobalt L$_{2,3}$-edge XAS spectra of M-CoPc-RT/400 and B-CoPc-RT/400, respectively. **b, e** 3d orbital diagrams of M-CoPc-RT/400 and B-CoPc-RT/400, respectively.

**c, f** Schematic showing the effect of thermal treatment on the coordination environment of M-CoPc-RT/400 and B-CoPc-RT/400, respectively.

inverted peaks, different from the positive peaks over M-CoPc-400 (Fig. 3c, d). Generally, inverted peaks in ATR-SEIRAS spectrum indicate consumption of adsorbed species as compared to the condition of background. Therefore, the inverted C−H bands over B-CoPc-400 suggest that under applied cathodic potential, the coverage of C−H species decreased as compared to the OCP condition, and the increasing of cathodic bias promoted the C−H species desorption from the B-CoPc-400. Time-dependent infrared spectroscopy was carried out to explore the stability of intermediates during the CORR over B-CoPc-400. *CO, *CHO, and *C-H$_x$ signals could remain intact throughout the measurement, highlighting the excellent durability of B-CoPc-400 (Figs. S15 and S16).

To further examine the relation between kinetics of CORR and proton-feeding, the kinetic isotope effect (KIE) of H/D was measured using H$_2$O and D$_2$O as the proton sources, which was performed at −0.6 V (vs. RHE)[30]. When D$_2$O was used to replace H$_2$O in 0.5 M KOH or 0.5 M K$_2$SO$_4$ electrolyte, the partial current density of methanol was significantly reduced for M-CoPc-400. As shown in Fig. 3f, compared to B-CoPc-400, M-CoPc-400 displays a much higher KIE value of 2.10 and 2.13 in KOH and K$_2$SO$_4$ solution, respectively, indicating that the CORR over M-CoPc-400 was limited by proton transfer, consistent with the *operando* ATR-SEIRAS results. From the *operando* ATR-SEIRAS and KIE results, it can be deduced that the CORR over M-CoPc-400 and B-CoPc-400 was rate-limited by the *CO/*CH$_x$O hydrogenation and methanol desorption step, respectively.

**Mechanistic insights into spin structure**

To dig out the origin of fast rate of hydrogenation over B-CoPc-400 in CORR to methanol, Co L$_{2,3}$-edge XAS measurements were conducted to differentiate the electronic structure between M-CoPc-400 and B-CoPc-400. Transition metal L$_{2,3}$-edge XAS spectra are sensitive to the change of oxidation state, spin state and/or orbital. Figure 4a, d shows the experimental and simulated Co L$_{2,3}$-edge XAS spectra of the samples, which can be divided into L$_2$ and L$_3$ regions, corresponding to the

transition from 2p$_{1/2}$ and 2p$_{3/2}$ levels to unoccupied 3d orbitals, respectively. As shown in Fig. 4a and Fig. S17a, b, the Co L$_{2,3}$-edge XAS spectrum of M-CoPc-RT and M-CoPc-400 are similar, which display the same characteristic features as that of CoPc molecule[35,36]. Three features appear in the energy region of Co L$_3$-edge, denoted as A$_1$, A$_2$ and A$_3$ as shown in Fig. 4a, d. The peak A$_1$ can be assigned to the transition from 2p$_{3/2}$ to the 3d$_{z^2}$ orbitals, while the peaks A$_2$ and A$_3$ are originated from the transition to 3d$_{x^2-y^2}$ orbitals[37]. The Co L$_{2,3}$-edge XAS spectrum of M-CoPc-RT/400 were simulated using the configuration interaction cluster model that includes the full atomic Multiplet theory and the hybridization with the ligands[38]. The Co 3d to N 2p transfer integrals were estimated for the various Co-N coordinations according to Harrison's prescription[39]. Calculations with the parameters (10Dq = 2.5 eV, Ds = 0.42 eV, Dt = 0.25 eV, Δ = 6.0 eV for M-CoPc-RT and 10Dq = 2.0 eV, Ds = 0.45 eV, Dt = 0.17 eV, Δ = 6.0 eV for M-CoPc-400) give rise to the theoretical spectra well consistent with the experimental results, as plotted in Fig. 4a. The ground states of Co$^{2+}$ in both M-CoPc-RT and M-CoPc-400 are in $^2$A$_{1g}$ ($d_{x^2-y^2}^{20}d_{z^2}^{21}d_{xy}^2d_{xz,yz}^4$) symmetry with a total spin of S = $^1/_2$. Figure 4b shows the 3d orbital diagrams of M-CoPc-RT and M-CoPc-400, constructed by the simulation parameters of Co L$_{2,3}$-edge XAS. Figure 4d shows the Co L$_{2,3}$-edge XAS spectrum of B-CoPc-RT, which is similar to that of M-CoPc-RT and M-CoPc-400, implying that the Co in B-CoPc-RT has the same ground state ($d_{x^2-y^2}^{20}d_{z^2}^{21}d_{xy}^2d_{xz,yz}^4$) with that of M-CoPc-RT/400, which is also in the low-spin state.

While, after heat treatment, the line shape of the L$_{2,3}$-edge of Co$^{2+}$ in B-CoPc-400 changed dramatically: L$_{2,3}$-edge shifted to lower binding energies and the contribution of transition from 2p to 3d$_{z^2}$ orbitals significantly decreased, similar to the high-spin Co$^{2+}$ in YBaCo$_3$AlO$_7$ with T$_d$ symmetry[40]. The theoretical spectra (Fig. 4d) were also calculated with the parameters (10Dq = 1.3 eV, Ds = 0.36 eV, Dt = 0.10 eV, Δ = 6.0 eV for B-CoPc-RT and 10Dq = 0.1 eV, Ds = 0.09 eV, Dt = −0.01 eV, Δ = 6.0 eV for B-CoPc-400), and the results (Fig. 4d and Fig. S17c, d) reveal that the Co in B-CoPc-RT was still at the ground state in $^2$A$_{1g}$ symmetry ($d_{x^2-y^2}^{20}d_{z^2}^{21}d_{xy}^2d_{xz,yz}^4$), however, with severe distortion

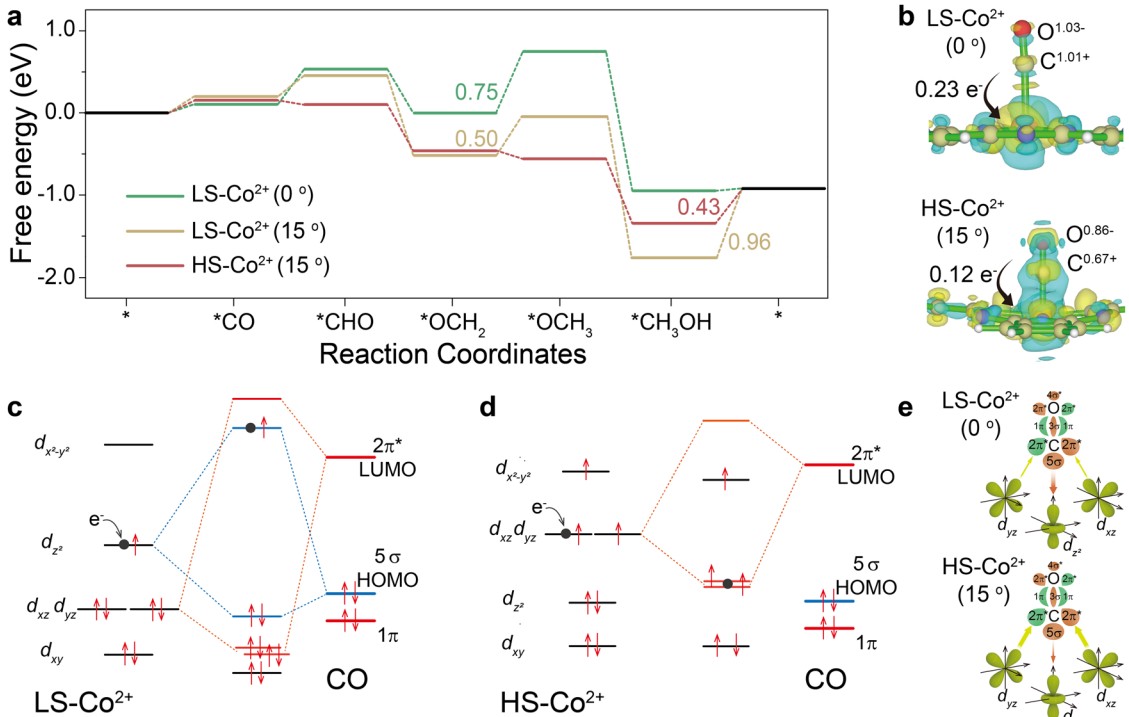

**Fig. 5 | CORR mechanism. a** Free energy diagram of CORR over LS-Co²⁺-(0 and 15°) and HS-Co²⁺-(15°). **b** The charge density distributions of LS-Co²⁺-CO and HS-Co²⁺-CO. Blue, yellow, pink and gray balls represent N, C, Co and O atoms, respectively, and the faint yellow and cyan regions refer to the increased and decreased charge density. Interactions between CO molecular frontier orbitals (5σ and 2π*) and the 3d orbital of **c** LS-Co²⁺ and **d** HS-Co²⁺ site. Dot in black of **c** and **d** is the electron from electrode under cathodic bias. **e** Schematic of σ and π-donation bonds between CO and 3d orbital of LS-Co²⁺ and HS-Co²⁺. The size of the arrow indicates the binding strength of 3$d_{z^2}$-5σ (in red color) and 3$d_{xz}/d_{yz}$-2π* (in green color), which also indicates the magnitude of the electron migration.

of coordination environment around Co after heat treatment, the 10Dq and $\Delta e_g$ reduced from 1.3 eV and 1.85 eV for B-CoPc-RT to 0.1 eV and 0.3 eV for B-CoPc-400 (Fig. S18b). As a result, the ground state of electron configuration changed from low-spin (b₁g ($d_{x^2-y^2}^{20}$), a₁g ($d_{z^2}^{21}$), eg ($d_{xz}^2$, $d_{yz}^2$), b₂g ($d_{xy}^2$)) to high-spin ((b₁g ($d_{x^2-y^2}^{21}$), eg ($d_{xz}^1$, $d_{yz}^1$), a₁g ($d_{z^2}^2$), b₂g ($d_{xy}^2$)) as shown in Fig. S18b and Fig. 4e, corresponding to a symmetry change from planar D4h to distorted D4h symmetry of Co²⁺. As shown in Fig. 4e, the transition from LS to HS of the Co center, accompanied with a decreased a₁g ($d_{z^2}^2$) and an increased eg ($d_{xz}^1$, $d_{yz}^1$) orbital level would affect the adsorption of CO molecule on the Co center as well as the charge distribution in *CO.

Electron paramagnetic resonance (EPR) measurements were further conducted to obtain information on the unpaired electron of Co in the catalyst (Fig. S19). The EPR spectrum of M-CoPc-RT and B-CoPc-RT show a strong signal at $g = 2.71$, which is a typical characteristic signal of low-spin Co²⁺, suggesting that unpaired electron exists in M-CoPc-RT and M-CoPc-RT. In addition, this signal still exists when M-CoPc-RT is annealed at 400 °C to obtain M-CoPc-400, indicating that Co in M-CoPc-400 is still at the low-spin state. However, when B-CoPc-RT is annealed at 400 °C to obtain B-CoPc-400, the signal at $g = 2.71$ reduces, and at the same time, a new signal appears at $g = 2.00$, which is a typical high-spin Co²⁺ signal. The EPR results suggest that the 400 °C treatment of B-CoPc-RT changes its spin structure from low spin to high spin, in agreement with the L$_{2,3}$-edge analysis.

### Theoretical calculation

DFT calculation was employed to study the influence of structure distortion of CoPc on spin state of Co²⁺. As shown in Fig. S20a, it is found that for planar CoPc molecule, the low spin state of CoPc (LS-Co²⁺) is more stable than the high spin state (HS-Co²⁺). However, if a distortion is applied (rotation degree from 0° to 30°), the energy of HS-Co significantly decreases, both HS and LS states become energetically

favorable, which indicates that the CoPc can be easily toggled back and forth between the two states.

To further clarify the effect of Co spin state on CORR performance, DFT calculation was performed to compare the reaction free energy of CORR on CoPc and distorted CoPc. Figure S20b displays the optimized structure of LS-Co²⁺ and HS-Co²⁺ with different rotation angles. Taking CoPc with rotation angles of 0° and 15° as a representative, there are two possible mechanisms of CORR to methanol over CoPc with rotation angles of 15° (LS-Co²⁺-(15°) and HS-Co²⁺-(15°)). As shown in Fig. 5a, compared to HS-Co²⁺-15°, the rate determining step of CORR over LS-Co²⁺-(15°) is the desorption of *CH₃OH corresponding to $\Delta G$ of 0.96 eV, 0.53 eV higher than that over HS-Co²⁺-15°, indicating energetically favorable H-Co²⁺-(15°) catalytic center for CORR to methanol. CO adsorption on LS-Co²⁺-0° is slightly energetically more favorable than that on LS-Co²⁺-15° and HS-Co²⁺-15°, while the CORR over LS-Co²⁺-0° is limited by the step of *CO and *OCH₂ hydrogenation, which become energetically favorable over HS-Co²⁺-15°. The rate-determining step (RDS) of CORR to CH₃OH changes from hydrogenation of *OCH₂ on LS-Co²⁺-0° to desorption of *CH₃OH on HS-Co²⁺-15°, corresponding to an energy requirement decrease by 0.32 eV (from 0.75 to 0.43 eV), in accordance with the *operando* ATR-SEIRAS and KIE results. Moreover, the latter corresponds to physical desorption without electron transfer, which is more likely to occur and can be achieved by many thermodynamic methods such as heating and stirring. The charge density difference calculations and corresponding charge transfer analysis (Fig. 5b) show that the net charge transfer ($n_{CT}$) from CO to Co²⁺ is 0.23e and 0.12e for LS-Co²⁺-0° and HS-Co²⁺-15°, respectively, and similar charge density distribution trend is observed for *CH₂O intermediate (Fig. S21). The $n_{CT}$ for CO adsorbed on Co site is the consonant contributions of both the σ and π components. Based on theoretical study[41], CO adsorption on Co site of LS/HS-CoPc is a two-step process, beginning with donation of electrons from the 5σ

orbital (HOMO: the highest occupied molecular orbital) of CO to the $3d_{z^2}$ orbital of Co, followed by back electron donation from the Co $d_{xz}/d_{yz}$ orbitals to the $2\pi^*$ orbital (LUMO: the lowest unoccupied molecular orbital) of CO.

The interactions between CO molecular frontier orbitals ($5\sigma$ and $2\pi^*$) and the 3d orbitals of LS-Co$^{2+}$-0° and HS-Co$^{2+}$-15° are displayed in Fig. 5c and d, respectively. As compared to LS-Co$^{2+}$-0°, $3d_{z^2}$ orbital of HS-Co$^{2+}$-15° is fully occupied, resulting in weakened CO adsorption over HS-Co$^{2+}$, consistent with the DFT calculation. On the other hand, compared to LS-Co$^{2+}$-0°, the two unpaired electrons in $3d_{xz}/d_{yz}$ orbitals of HS-Co$^{2+}$-15° are more active than those in $3d_{xz}/d_{yz}$ orbitals of LS-Co$^{2+}$-0°, which can easily back-donate to the $2\pi^*$ of CO (p orbital of C and O) as formation of HS-Co$^{2+}$-15°-CO. Moreover, due to anti-bonding feature of the $3d_{xz}/d_{yz}$-$2\pi^*$ orbital, the $3d_{xz}/d_{yz}$-$2\pi^*$ bonding will weaken the C−O bond in *CO, which can facilitate the following step of *CO hydrogenation. Figure 5c, d shows the schematic of σ and π-donation bonds between CO and 3d orbitals of LS-Co$^{2+}$-0° and HS-Co$^{2+}$-15°; the weaker $3d_{z^2}$-$5\sigma$ and more electron transfer from Co to *CO via π back-donation for HS-Co$^{2+}$-15° lead to smaller $n_{CT}$ from CO to Co site, enabling more electron accumulation on the $2\pi^*$ orbital of *CO, which will effectively promote the hydrogenation of CO reduction intermediates ca. *CO (similar to *OCH$_2$) in CORR. Once *CO formation on Co sites occurs, under cathodic bias, the electrons from the electrode will fill into the highest unpaired orbitals of Co sites, $3d_{z^2}$ and $3d_{xz}/d_{yz}$ for LS-Co$^{2+}$-0° and HS-Co$^{2+}$-15°, respectively. In the case of LS-Co$^{2+}$-0°-CO, the electrons from the electrode will fill in the anti-bonding orbital of $3d_{z^2}$-$5\sigma$ bond, resulting in weakened *CO adsorption strength, unfavorable for *CO reduction. While in the case of HS-Co$^{2+}$-15°-CO, the electrons from the electrode will fill in the bonding orbital of $3d_{xz}/d_{yz}$-$2\pi^*$ bond, weakening the C−O bond in *CO, which can effectively promote *CO hydrogenation. In addition, DFT calculations also show that H$^+$ adsorption on Co$^{2+}$ of HS-Co$^{2+}$-15° requires an energy of 0.68 eV, much larger than that over LS-Co$^{2+}$-0° (0.44 eV) (Fig. S22), which shall further benefit CORR against hydrogen evolution reaction (HER).

## Discussion

To sum up, L-edge XAS analysis on a model single-Co-atom catalyst verified the transformation of single atomic cobalt center from LS (S = 1/2) to HS (S = 3/2) after thermal treatment, which greatly promoted CORR performance. Combining *operando* ATR-SEIRAS, KIE experiments and DFT calculations, it showed that the change of spin state of single atomic cobalt center (II) from LS (S = 1/2) to HS (S = 3/2) resulted in a change of the CORR RDS from *CO/*CH$_x$O hydrogenation to methanol desorption, corresponding to a decrease of the RDS energy barrier by 0.32 eV from 0.75 to 0.43 eV. Our work indicates that the chemical bonding of CO molecule on single atom catalyst determined by the specific orbital characters of d-bands is critical for the hydrogenation process in CORR, which provides valuable information for understanding the electrochemical catalytic reaction at the molecular level.

## Methods
### Chemicals
Cobalt phthalocyanine (M-CoPc) and binuclear cobalt phthalocyanine (D-CoPc) were purchased from Sigma-Aldrich and used directly without further purification. Melamine and L-alanine were purchased from Shanghai Aladdin Biochemical Technology Co., Ltd., Ethanol, dimethylformamide, and hydrochloric acid were obtained from Alfa Aesar (China) Chemicals Co., Ltd. Nafion™ 115 solution (5 wt.% in lower aliphatic alcohols and water) was purchased from Sigma-Aldrich and used directly. Carbon black (XFP05 7440-74-0) was purchased from XFNANO Materials Tech Co., Ltd. Deionized water from Millipore Q water purification system was used to prepare all solutions.

### Synthesis of nitrogen-doped carbon (NC)
In a typical synthesis, a mixture of melamine (C$_3$H$_6$N$_6$) (8 g) and L-alanine (C$_3$H$_7$NO$_2$) (1.5 g) was first ground into a homogeneous precursor using zirconia ball in a nylon jar. Subsequently, the fine powder mixture underwent a two-stage pyrolysis and carbonization process (first stage: from 25 to 600 °C at a ramping rate of 3 °C/min, maintain at 600 °C for 2 h; second stage: from 600 to 900 °C at a ramping rate of 2 °C/min, maintain at 900 °C for 1 h) in a tubular furnace (Carbolite, UK) in argon atmosphere. After cooling down to room temperature, the product was successively leached at 80 °C in 1 M HCl for 12 h. Afterwards, the sample was heated again at 850 °C in Ar for 1 h to recover the crystallinity of carbon.

### Synthesis of M-CoPc-RT and B-CoPc-RT
In a typical experiment, 100 mg of NC was dispersed in 100 mL of DMF under sonication for 1 h, followed by adding 5 mg of M-CoPc dispersed in DMF. After 30 min of sonication, the mixture was stirred for 48 h at room temperature. The precipitate was then collected by vacuum filtration and sufficiently washed by DMF and ethanol until colorless. Finally, the precipitate was lyophilized to yield the M-CoPc-RT. B-CoPc-RT was prepared using the same method by replacing M-CoPc with B-CoPc.

### Synthesis of M-CoPc-400 and B-CoPc-400
The as-prepared M-CoPc-RT/B-CoPc-RT was thermally treated at 400 °C for 2 h in a tubular furnace in argon atmosphere to prepare M-CoPc-400/B-CoPc-400.

### Characterizations
X-ray diffraction (XRD) patterns were recorded on a Bruker D8 diffractometer in the $2\theta$ range from 10 to 80° using Cu Kα radiation ($\lambda = 1.5406$ Å). Nitrogen adsorption-desorption isotherms were measured at 77 K on a Micromeritics Tristar 2420 analyzer (USA). Before measurements, the samples were degassed in vacuum at 190 °C for 10 h. The Brunauer-Emmett-Teller (BET) and Barrett-Joyner-Halenda (BJH) methods were used to calculate the specific surface area, pore volume, and pore size distribution, respectively. Field-emission scanning electron microscopy (FE-SEM) images were taken on a ZEISS Merlin Compact SEM equipped with energy-dispersive X-ray spectroscopy (INCAPentalFETx3 Oxford EDS). Sub-ångström-resolution high-angle annular dark-field scanning transmission electron microscopy (HAADF-STEM) characterization was performed on a JEOL JEMARM200F STEM/TEM with a guaranteed resolution of 0.08 nm. Chemical states and composition of the samples were analyzed by X-ray photoelectron spectroscopy (XPS) on an ESCALAB 250 photoelectron spectrometer (Thermo Fisher Scientific) using a monochromatic Al Kα X-ray beam (1486.6 eV). All binding energies were referenced to the C 1s peak (284.6 eV). Inductively coupled plasma atomic emission spectroscopy (ICP-AES) measurements were conducted on Atomscan Advantage, Thermo Jarrell Ash, USA.

### Electrochemical measurements
All electrochemical measurements remain consistent with our previous methods[20,28]. All electrochemical measurements were carried out at ambient temperature and pressure on a rotating disc electrode system (Pine Inc.) by a CHI 760e potentiostat. A three-electrode cell configuration was employed with a working electrode of glassy carbon rotating disc electrode (RDE) of 5 mm diameter, a counter electrode of graphite rod (5 mm diameter), and a saturated calomel reference electrode (SCE). The saturated calomel reference electrode ($E_{RHE} = E_{SCE} + 0.0591 \times pH + 0.244$) was calibrated before the electrochemical measurement (Figure S23). Figure S24 shows the ohmic drop during CORR. To prepare the catalyst ink, 5 mg of catalyst and 25 µL of 5 wt.% Nafion™ solution (DuPont) were introduced into 975 µL of water-isopropanol solution with equal volume of water and

isopropanol and sonicated for 3 h. An aliquot of 6 μL of the catalyst ink was applied onto a glassy carbon RDE and allowed to dry naturally in air, giving a catalyst loading of 0.15 mg/cm². A CO-saturated electrolyte was prepared by purging CO (99.99%) into 0.5 M KOH aqueous solution for 20 min, and a flow (100 cc/min) of CO was maintained over the electrolyte throughout the electrochemical measurements. For comparison, CV measurements were also performed in Ar (99.99%)-saturated electrolyte. All potentials were calculated with respect to the reversible hydrogen electrode (RHE) scale according to the Nernst equation. The pH was determined by a pH meter (S220 SevenCompact™ pH/Ion). The linear sweep voltammograms (LSV) were collected at a scan rate of 5 mV/s. All LSV data were corrected for the Ohmic drop. All current densities were normalized to the geometric area of the electrode.

## Electrochemical flow cell for CO reduction
Electrochemical CO reduction was carried out in a flow cell. The windows for electrolysis were set to 2 cm × 2 cm. Each chamber has an inlet and an outlet for electrolyte, and an RHE reference electrode was placed in the catholyte chamber. The catalyst ink was prepared by mixing 10 mg of catalyst, 10 mL of ethanol, and 200 μL of a Nafion perfluorinated resin solution. Then, catalysts were air-brushed onto 2 × 2 cm² gas diffusion layer (Fuel Cell Store) electrodes with mass loading of 1.0 mg/cm², and used as the cathode. A 2 × 2 cm² Ni foam was used as a counter electrode for oxygen evolution reaction. An anion exchange membrane (Dioxide Materials) was used to separate the cathode and anode chambers. 0.5 M KOH solution was used as the electrolytes. The catholyte and anolyte were cycled at a flow rate of 80 mL min⁻¹ by using a peristaltic pump. The gas inlet and outlet on the cathode side were linked to a CO gas-flow meter (5 sccm) and a GC, respectively. The gaseous products (i.e., $H_2$ and $CH_4$) were quantified by a gas chromatography (GC-2014, SHIMADZU) equipped with a flame ionization detector (FID) for $CH_4$ and a thermal conductivity detector (TCD) for $H_2$ quantification. Ultra-pure helium (He, 99.9999%) was used as the carrier gas. The flow rate of CO was controlled at 5 sccm/min at the inlet of electrochemical cell by a standard series mass flow controller (Alicat Scientific mc-5 sccm) and measured by a flow meter (Thermo Scientific GFM Pro) at the exit of the electrochemical cell. GC was calibrated using standard gas mixtures under standard conditions (1 atm and 298 K). The liquid product ($CH_3OH$) was characterized by hydrogen nuclear magnetic resonance (¹H NMR) after the entire electrochemical measurement on a Bruker Avance III 600 MHz nuclear magnetic resonance spectrometer (Germany) with dimethyl sulfoxide (DMSO) as the internal standard.

## Faradaic efficiency (FE) quantification
Based on the definition of Faradaic efficiency:

$$FE_i = \frac{Q_i}{Q_{total}} \tag{1}$$

where $i$ represents $CH_3OH$, $H_2$, or $CH_4$.

$Q_i$ and $Q_{total}$ can be obtained from the following equations:

$$FE_i = Z_i \times F \times N_i \tag{2}$$

$$Q_{total} = I \times t \tag{3}$$

Based on the GC data and ideal gas law:

$$N_i = N_{total} \times V_i \tag{4}$$

$$N_{total} = \frac{P_0 \times V_0}{R \times T_0} \tag{5}$$

$$V_0 = G \times t \tag{6}$$

where $G$ is the volumetric flow rate.

Thus, Eq. (1) can be written as:

$$FE_i = \frac{z_i \times v_i \times G \times F \times P_0}{I \times R \times T_0 \times 60000} \tag{7}$$

where $N_i$: mole of product $i$ in the GC sampling loop; $N_{total}$: mole of all gases in the GC sampling loop, $V_0$: volume of the GC sampling loop, $t$: time for gas to fill the GC sampling loop, $I$: the average current in a period ($t$) of electrocatalysis:

$$I = \frac{\int_0^t J(t)dt}{t} \tag{8}$$

$V_i$: the volume ratio of product $i$ in the GC sampling loop, $Z_i$: number of electrons required to produce an $i$ molecule, which is 4, 2, and 6 for $CH_3OH$, $H_2$, and $CH_4$, respectively, $F$: Faradaic constant (96485 C/mol), $P_0$: atmospheric pressure $1.013 \times 105$ Pa, $T$: reaction temperature at 298 K, $R$: ideal gas constant, 8.314 J·mol·K⁻¹. Turnover frequency (TOF) of methanol was calculated as follows:

$$TOF = \frac{I_{product}/NF}{m_{cat} \times w/M_{metal}} \times 3600 \tag{9}$$

$I_{product}$: partial current for methanol, $N$: number of electrons required to produce a methanol molecule, which is 4, $F$: Faradaic constant (96485 C/mol), $m_{cat}$: mass of catalyst on the electrode, g, $w$: metal loading in the catalyst based on ICP-AES results.

## X-ray absorption spectroscopy
X-ray absorption spectroscopy (XAS) was collected by employing synchrotron radiation light source at BL14W1 beamline of Shanghai Synchrotron Radiation Facility (SSRF) at room temperature. Energy calibration was performed with a Co foil standard by shifting all spectra to a glitch in the incident intensity. Fluorescence spectra were recorded using a seven-element Ge solid state detector. The XAFS data were recorded in fluorescence excitation mode using a Lytle detector, and the spectra of all references were collected in the transmission mode. All samples were pelletized as disks of 13 mm diameter and 1 mm thickness using boron nitride as the binder. The acquired XAFS data were analyzed by Athena and Artemis software, according to the standard procedures. The k³-weighted EXAFS spectra were obtained by subtracting the post-edge background from the overall absorption and then normalized with respect to the edge-jump step. Subsequently, χ (k) data of Co K-edge in the k-space from 2.7 to 11.1 Å⁻¹ were Fourier transformed to real (R) space using a hanning windows (dk = 1.0 Å⁻¹) to separate the EXAFS contributions from different coordination shells. To obtain the quantitative structural parameters around Co atoms, least-squares curve parameter fitting was performed. Effective scattering amplitudes and phase-shifts for the M-CoPc-RT, M-CoPc-400, B-CoPc-RT, and B-CoPc-400 were calculated using the ab initio code FEFF 8.2[42]. The amplitude reduction factor $S_0^2$ was obtained by fitting Co foil and $Co_3O_4$. In the subsequent fitting of M-CoPc-RT, M-CoPc-400, B-CoPc-RT and B-CoPc-400, the coordination number N, interatomic distance R, Debye-Waller factor $\sigma^2$ and the edge-energy shift $\Delta E_0$ were allowed to run freely.

## Simulation method of Co L-edge

The simulation of Co $L_{2,3}$-edge XAS spectra were performed based on the configuration interaction cluster model[38], Hartree-Fock estimates of the radial part of the Coulomb interaction in terms of Slater integrals $F^k$ and $G^k$ and the spin-orbit coupling parameters $\zeta(3d)$ and $\zeta(2p)$ using the atomic theory developed by Cowan[43]. The reduction of 70% for Slater integrals was applied to account for the over-estimation of electron-electron repulsion found in the calculations of free ion. The hybridization of Co 3d orbit and N 2p orbit was treated using a charge-transfer model, in which $3d^n$ and $3d^{n+1}\underline{L}$ configuration were involved. The energy difference of these configurations is defined using the charge-transfer energy $\Delta$ and the 3d-3d Coulomb interaction $U_{dd}$ (6.5 eV). The ground states $\varphi$ can be written as $\varphi = \alpha_i|3d^n\rangle + \beta_i|3d^{n+1}\underline{L}\rangle$.

## *Operando* ATR-SEIRAS

The attenuated total reflectance surface-enhanced infrared absorption spectroscopy (ATR-SEIRAS) experiments were performed on a Nicolet iS50 FT-IR spectrometer equipped with a MCT detector cooled with liquid nitrogen and PIKE VeeMAX III variable angle ATR sampling accessory.

## Theoretical methods and computational details

All DFT calculations were performed in the standard Gaussian 16 software package[44]. Following which, the calculation of geometry optimization was carried out by using the PBE-D3(BJ)[45,46] (including London-dispersion correction) function with the 6−31 G(d) basis set for all atoms. The vibrational frequency calculation was subsequently carried out at the same level as geometry optimization for two purposes: (1) characterizing the nature of the stationary point that all frequencies of local minimum is positive; (2) obtaining the thermodynamic quantities of the studied species at 298 K and 0.1 MPa such as relative Gibbs free-energies. In addition, the single-point energy of studied stationary point was calculated by using the PBE-D3(BJ) function with the 6−311 G(d,p) basis set for all atoms.

## Data availability

All data are reported in the main text and supplementary materials. Source data are provided with this paper. All relevant data are available from the authors on reasonable request. Source data are provided with this paper.

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

## Acknowledgements

We acknowledge funding support from CAS Project for Young Scientists in Basic Research (YSBR-022), the Strategic Priority Research Program of the Chinese Academy of Sciences (XDB36030200), the National Natural Science Foundation of China (Grant No. 22075195, 21974103, 22102207, 2199152 and 21832004), the start-up funds of Wuhan University, the Singapore Ministry of Education Academic Research Fund (AcRF) Tier 1: RG4/20 and RG2/21, Tier 2: MOET2EP10120-0002, Agency for Science, Technology and Research: AME IRG A20E5c0080 and Photon Science Center for Carbon Neutrality.

## Author contributions

J.D., H.Y., Y.Z., and B.L. conceptualized the project. H.Y., Y.H., Y.Z., and B.L. supervised the project. J.D. and Z.W. synthesized the catalysts, conducted the catalytic tests with the help of Q.Z. and the related data processing with the help of J.Zha., Y.L., W.W., W.L. Z.Z., R.Y., and X.S., as well as performed the materials characterization and analysis with the help of X.S., J.Zha., and C.S. J.D. conducted the in situ ATR-SEIRAS measurements with the help of Z.W. J.Zho. performed the L-edge analysis. F.L. carried out the theoretical study. J.D., Z.W., F.L., Y.Z., and B.L. wrote the manuscript with support from all authors.

## Competing interests

The authors declare no competing interests.
