## [Peer Review File · Nature Communications]

Atomic High-Spin Cobalt(II) Center for Highly Selective Electrochemical CO Reduction to CH₃OHREVIEWER COMMENTS

Reviewer #1 (Remarks to the Author):

In this manuscript, Ding and coworkers presented a work in which CO was electroreduced to methanol efficiently with a high Faradaic efficiency of 57%. To achieve this performance, the authors developed a high spin binuclear CoPc catalyst which was proposed to have the different mechanism for methanol production compared to the typical low spin mononuclear CoPc. Though this finding is of interest for the electrocatalytic community, the reviewer thinks the study and the presented data are not comprehensive and compelling enough. Some questions and concerns need to be addressed before the manuscript can be assessed for publication.

1. Additional isotopic experiments should be supplemented. ^{13}CO electrolysis should be conducted to confirm the produced methanol comes from the CO reduction rather than the other possible carbon sources like the carbon support and any contaminations. The authors attributed the peak at $\sim 1400\text{ cm}^{-1}$ to $^*\text{CHO}$ and it was proposed to be the important intermediate in methanol production. It's necessary to be further demonstrated by replacing H_2O with D_2O and see if this peak will shift.
2. Data inconsistency. The total current density calculated based on the partial current density of methanol and its Faradaic efficiency in Fig.2c is very different from the data shown in Fig. 2d at the same conditions.
3. The authors attributed the improved methanol formation on high spin CoPc to the higher binding strength of CO. As the only experimental evidence, the SEIRAS results show that the $^*\text{CO}$ vibrational frequency on high spin CoPc is lower than that on low spin CoPc, which is completely contrary to what they claimed because higher $^*\text{CO}$ vibrational frequency means lower CO binding strength.
4. The reviewer cannot get the meaning of spending such a great length of the manuscript to discuss the inverted C-H bonds on page 8. How did the authors assign the C-H peak? Is it possible that the C-H species comes from the organic contaminants since it seems irrelevant to the methanol production?
5. The high spin state of Co was achieved after thermal treatment. It seems that the temperature has a drastic impact. Did the authors compare the performance of catalysts treated with different temperatures?
6. Why did the authors use the nitrogen-doped carbon as the substrate? What if you use other typical carbon substrates such as Vulcan carbon black and graphene?
7. On page 8, line 177, the authors claimed that the CORR over M-CoPc-400 was limited by proton transfer based on the KIE results and thought it was consistent with the operando ATR-SEIRAS results. However, the reviewer didn't see any corresponding results from SEIRAS study.
8. The related NMR spectra of the resulting electrolytes after CO electrolysis and KIE experiments should be provided.

Reviewer #2 (Remarks to the Author):

In this manuscript, Ding et al. engineered the conformation of the cobalt active site within cobalt phthalocyanine (CoPc) molecular catalyst, and achieved an improved performance for CO

electroreduction to methanol. Specifically, the authors suggested that the catalytic activity of CoPc can be manipulated by tuning the spin-state of the Co 3d electrons. When the electron rearrangement of the Co 3d orbitals of CoPc change from the low-spin state to the high-spin state, the catalyst TOF was observed to increase by 5 times. Besides, the ATR-SEIRAS and DFT studies all supported this argument. Overall, I support the publication of this manuscript in Nature Communication provided the authors can address the following minor issues.

Comments:

(1) While the M-CoPc and D-CoPc were purchased from chemical vendors, it is suggested to carry out some basic physical characterizations on them to confirm their chemical structures and purity, i.e., MS, NMR.

(2) It seems that the authors have measured the Co-loading of M-Co-Pc and D-CoPc on nitrogen doped carbon, before and after the annealing, for estimating the TOF, however, these data are not available. Please provide these data and discuss whether the Co-loading plays a role in the catalysis.

(3) It is a bit surprise to see that methanol formation on B-CoPc-400 is limited by the desorption step. If it is the case, shouldn't the authors see more C-based species from ATR-IR, e.g., *CO, *C-H species, for the sample of B-CoPc-400, which is not the case. Besides, the authors attributed the "inverted C-H peaks" on B-CoPc-400 to the "consumption of adsorbed species as compared to the condition of OCP", this is hard to believe, one would not expect the formation of C-H species at OCP conditions. Moreover, as shown in Fig. S8, these "inverted C-H peaks" already exist in absence of CO in Ar-saturated 0.5 M KOH.

Reviewer #3 (Remarks to the Author):

The study, entitled "Atomic High-Spin Co (II) Center for Highly Selective Electrochemical CO Reduction to CH₃OH," by Ding et al., reported a high-spin state binuclear cobalt phthalocyanine on nitrogen-doped carbon support (B-CoPc-400), which exhibits superior performance in the electroreduction of CO to CH₃OH, with a partial current density of 35 mA cm⁻² and a faradaic efficiency of 50% at -0.8 V vs RHE. Through the combination of Operando measurements and density functional theory calculations, the authors uncovered the mechanism by which the high-spin state of Co²⁺ weak the C-O bonding through electron accumulation on *CO, thereby improving its CORR performance. The paper is well-written and provides a comprehensive understanding of energy conversion through molecular catalysis. Thus, the manuscript is recommended for publication in Nature Communications after addressed the following questions.

1. In line 26, the term "RHE" (reversible hydrogen electrode) should be introduced and subsequently used as an abbreviation throughout the manuscript.
2. In lines 51-52, the claim that "Cobalt phthalocyanine (CoPc) molecular catalyst showed the capability

to electrochemically reduce CO to produce CH₃OH" is supported by references 19-21, which however pertain to the electrochemical reduction of CO₂. It is suggested that the author carefully check and replace these references with more appropriate ones that support the claims made.

3. The in situ Raman spectrums of M-CoPc-RT and B-CoPc-RT during thermal treatment, as presented in Fig. 1b and Fig. 1c, respectively, both show a weakened intensity of the D and G band at 400 oC. An explanation of this observation should be provided by the authors.

4. The FT-EXAFS spectra of B-CoPc-400 reveal new peaks at ~1.7 Å and ~2.2 Å in comparison to B-CoPc-RT, while M-CoPc-400 does not show significant differences from M-CoPc-RT. Further explanation is suggested.

5. The reference potentials in the manuscript should be presented consistently, either with or without brackets. (e.g., line 131 and line 134)

6. A comparison of the performance with recently reported articles would be beneficial. Thus, a table summarizing this comparison could be included in the supplementary information.

7. The system in which the LSV curves were measured, whether a flow cell or a typical three-electrode cell, should be specified. The reason for measuring the curves on the carbon paper instead of the GDE or glassy carbon rotating disc electrode should also be discussed.

8. The authors should confirm whether the saturated calomel reference electrode was calibrated before the electrocatalytic performance test.

9. The total FE shown in Fig. 2c do not appear to be 100%. It would be useful to provide a table of graph showing all the product, including any other gas/liquid products that may have been generated.

10. The authors are suggested to include ¹H NMR graphs for Fig. 2c.

11. Have the authors measured the CORR performance of B-CoPc-RT? Otherwise, it is hard to say that high-spin state B-CoPc (B-CoPc-400) can catalyze CORR to methanol much more effectively than low-spin state B-CoPc (B-CoPc-RT).

12. Characterization of the samples after the stability test, such as HRTEM and XAS, would be valuable to confirm whether the Co retains its atomic dispersion after the CORR.

13. To improve the comprehensiveness, these close literature should be cited: Chem Catalysis 2, 372-385 (2022); Angew. Chem. Int. Ed. 61, e202110186 (2022). Adv. Mater. 34, 2106541 (2022).

14. It appears that some of the figure numbers in the manuscript may be incorrect. For example, in line 202, it should be Fig. 4a instead of Fig. 3a. The same applies to lines 203, 205, 211, and 219. The figure number in line 213 is missing. The authors are advised to carefully check all the figure numbers.

15. The style of describing figures in the manuscript should be standardized, either as Fig. xx or Figure xx.

16. In line 343, the authors state that all LSV data were corrected for the ohmic drop. The accurate value of the ohmic drop, as well as the method used to measure it, should be reported.

Reviewer #4 (Remarks to the Author):

Recommendation: Publish after major revisions noted.

In this study, the author designed the structure of Co active center in cobalt phthalocyanine (CoPc) molecular catalyst, and obtained the B-CoPc-RT/400 catalyst with significantly optimized catalytic performance of electrochemical carbon monoxide (CO) reduction to methanol. Through a series of experimental characterization and density functional theory (DFT) calculations, it is shown that the Co 3d orbitals of CoPc transferred from the low-spin state (LS, 1/2) to the high-spin state (HS, 3/2), induced by the electron rearrangement Co for the change of molecular conformation, which weakens the C-O bond in *CO, promotes the hydrogenation of the CORR intermediate, and is responsible for greatly enhancing the CORR performance. However, this study has methodological problems and limited innovation, thus the paper is recommended to be major revised. The detailed comments are as follows:

1. The study constructed cobalt phthalocyanine and binuclear cobalt phthalocyanine (M-CoPc and B-CoPc) catalyst anchored on nitrogen doped carbon support, while the reason to choose the binuclear molecule and monoatomic Co catalyst to study is not mentioned, which can be added in the introduction.
2. In DFT calculation, the authors believed that the spin state transition was caused by geometric distortion of CoPc. However, there is no any experimental evidences that support the structural distortion of CoPc.
3. From the results of the spin state change in the paper, it can be seen that the spin state of Co²⁺ changes from L.S. to H.S. after the transformation of B-CoPc-RT into B-CoPc-400 after heat treatment, while the spin state of M-CoPc-RT does not change significantly before and after heat treatment. Please clarify that the structure distortion of binuclear molecules and the heat treatment are the driving force of the spin state change. (Journal of Materials Chemistry A 2019, 7 (47), 27175-27185; Journal of the American Chemical Society 2018, 140 (45), 15149-15152; Energy & Environmental Science 2022, 15 (4), 1601-1610; Nature Communications 2021, 12 (1), 1734).
4. Please give other characterization verification of binuclear molecule structure of B-CoPc-RT/400 in addition to HAADF-STEM images.
5. The transition of spin state of Co²⁺ should be also characterized by other characterization techniques to measure the magnetic property of the catalysts, such as the magnetic susceptibility.
6. In Table S1. , the CN of Co-N in B-CoPc-400 is significantly smaller than B-CoPc-RT, and even M-CoPc-400, which is irrational, please give the explanation.
7. In Fig 2 and Fig 3, the comparison of catalytic performance and the discussion of CORR mechanism are mainly for M-CoPc-400 and B-CoPc-400, while Fig 4 points out that the spin-state transition of B-CoPc occurs before and after heat treatment. Please clarify the rationality of the performance comparison sample.
8. Please explain in detail how to obtain KIE values of different catalysts and the meaning of these values.
9. In line 153-155, the author claimed that "The *CO vibrational frequency on M-CoPc-400 is slightly lower than that on B-CoPc-400 (1911 and 1951 cm⁻¹ at 0 V vs. RHE), indicating a slightly higher binding strength of B-CoPc-400 towards CO.". May the authors provide evidence or literature reports to support this conclusion. Furthermore, in line 258-260, it was proposed by DFT study that "As compared to LS-Co²⁺, 3d_{z²} orbital of HS-Co²⁺ is fully occupied, resulting in weakened CO adsorption over HS-Co²⁺, consistent with the DFT calculation." Is that contradicting to the results of operando ATR-SEIRAS

measurement?

10. In line 210-216, "The theoretical spectra (Fig. 3d) were also calculated with the parameters ($10Dq = 1.3$ eV, $Ds = 0.36$ eV, $Dt = 0.10$ eV, $\Delta = 6.0$ eV for B-CoPc-RT and $10Dq = 0.1$ eV, $Ds = 0.09$ eV, $Dt = -0.01$ eV, $\Delta = 6.0$ eV for M-CoPc-400), and the results (Fig. d and Fig. S11c, d) reveal that the Co in B-CoPc-RT was still at the ground state in $2A_{1g}$ symmetry ($dx^2-y^2dz^2, dxz, yz^4$), however, with severe distortion of coordination environment around Co after heat treatment, the $10Dq$ and Δ_{eg} reduced from 1.3 eV and 1.85 eV for M-CoPc-RT to 0.1 eV and 0.3 eV for B-CoPc-RT (Fig. S12b)."

1) Please confirm: is M-CoPc-400 here correct? Because there is no spectra for M-CoPc-400 in Figure 3d.

2) Please confirm: is M-CoPc-RT here correct? $10Dq = 1.3$ eV for B-CoPc-RT.

3) Generally, $10Dq$ is called as crystal field splitting energy. Why the $10Dq$ is only 0.1 eV for B-CoPc-400 (if correct)?

11. Please pay attention to the one-to-one correspondence between the caption and the figure in the text and the accuracy of language expression, for example, Fig 3d appeared twice in the caption.

Response to Reviewers

We are very grateful to the critical comments and constructive suggestions provided by the reviewers, which shall significantly help to improve the quality of our work. Our manuscript has been revised accordingly, and the changes are highlighted by yellow colour in the text. The following list our responses to the comments from the reviewers. All the original comments are given in blue colour (italic) and our responses in black colour.

Reviewer #1 (Remarks to the Author):

In this manuscript, Ding and coworkers presented a work in which CO was electro-reduced to methanol efficiently with a high Faradaic efficiency of 57%. To achieve this performance, the authors developed a high spin binuclear CoPc catalyst which was proposed to have the different mechanism for methanol production compared to the typical low spin mononuclear CoPc. Though this finding is of interest for the electrocatalytic community, the reviewer thinks the study and the presented data are not comprehensive and compelling enough. Some questions and concerns need to be addressed before the manuscript can be assessed for publication.

Response: We sincerely appreciate the reviewer for the time and efforts spent in assessing our work. To fully address the reviewer's concerns, detailed responses and supporting data are provided below.

1. Additional isotopic experiments should be supplemented. ^{13}CO electrolysis should be conducted to confirm the produced methanol comes from the CO reduction rather than the other possible carbon sources like the carbon support and any contaminations. The authors attributed the peak at $\sim 1400\text{ cm}^{-1}$ to $^\text{CHO}$ and it was proposed to be the important intermediate in methanol production. It's necessary to be further demonstrated by replacing H_2O with D_2O and see if this peak will shift.*

Response: We thank the reviewer for the insightful suggestions. ^{13}CO electrolysis was carried out and the results are shown in **Figure R1-1a-c**. When ^{13}CO was used as the reactant, the CH_3OH H-NMR spectrum was clearly split, directly proving that the

product CH₃OH was derived from CO instead of other possible carbon sources. Meanwhile, the use of CO or ¹³CO did not change the Faradaic efficiency of CH₃OH (**Figure R1-1c**). Moreover, H₂O, D₂O and H₂O + D₂O were used in *operando* ATR-SEIRAS measurements. As shown in **Figure R1-1d-e**, when D₂O was used instead of H₂O, the position of CHO moved by 25 cm⁻¹ towards smaller wavenumbers due to isotopic redshift caused by the mass effect. Thus, it is confirmed that the peak at ~1400 cm⁻¹ can be ascribed to the *CHO intermediate in CO₂-to-methanol reduction.

Figure R1-1. ¹H 1D NMR spectra of products and DMSO recorded at different potentials (vs. RHE) over M-CoPc-400 (a) and B-CoPc-400 (b). c, Comparison of Faradaic efficiency of CH₃OH with ¹³CO and ¹²CO as the reactant. *Operando* ATR-SEIRAS spectra recorded over B-CoPc-400 in CO-saturated 0.5 M KHCO₃ (d. H₂O, e. D₂O, f. H₂O + D₂O).

2. Data inconsistency. The total current density calculated based on the partial current density of methanol and its Faradaic efficiency in Fig.2c is very different from the data shown in Fig. 2d at the same conditions.

Response: We thank the reviewer for pointing out the inconsistent data. We apologize that we used the wrong electrode geometry for our calculations. According to the reviewer's comment, the catalyst's stability was evaluated again, and the results are

shown in **Figure R1-2**. The experimental results show that the B-CoPc-400 could produce CH₃OH with a FE of 53% at -0.7 V vs. RHE with negligible current loss for 10 h, corroborating its excellent catalytic durability.

Figure R1-2. Stability of B-CoPc-400 recorded at -0.7 V (vs. RHE). The measurement was performed at the condition of 1 atm CO and room temperature in CO-saturated 0.5 M KOH.

*3. The authors attributed the improved methanol formation on high spin CoPc to the higher binding strength of CO. As the only experimental evidence, the SEIRAS results show that the *CO vibrational frequency on high spin CoPc is lower than that on low spin CoPc, which is completely contrary to what they claimed because higher *CO vibrational frequency means lower CO binding strength.*

Response: We thank the reviewer for raising the critical comment. According to the molecular orbital theory, the highest occupied molecular orbital (HOMO) of CO is the almost non-bonding σ orbital that is localized to C, and the lowest unoccupied molecular orbital (LUMO) is the antibonding π orbital. The involvement in formation of σ and π orbital are C and O atoms. **Figure R1-3a** displays the molecular orbital energy level of CO molecule, and the ground state configuration is: $1\sigma^2 2\sigma^2 1\pi^4 3\sigma^2$. The 1σ orbital is primarily localized on O, thus, they are essentially non-bonding or weakly bonding orbitals. 2σ orbital is bonding orbital. The 1π orbital is a double degenerate pair of π bonding orbitals, which mainly has the properties of a C_{2p} orbital. The HOMO

of CO is a 3σ orbital, which has the properties of a C_{2p_z} orbital, basically a non-bonding orbital, and is localized to C. The LUMO is a double degenerate pair of antibonding π orbitals, which has the properties of a C_{2p} orbital (**Figure R1-3b**). This combination of frontier orbitals (a pair of empty π orbitals that are essentially localized to carbon's fully filled σ orbitals) is the reason to form M_n -CO. CO is a ligand with a variety of coordination modes (**Figure R1-3c**), usually involving bridging one, two, or three metal atoms, and the expansion frequency of CO follows the following sequence: $MCO > M_2CO > M_3CO$. The coordination information shows that CO has excellent adsorption on the surface of catalysts. From previous studies, it can be seen that the CO adsorption energy and the C-O vibration frequency are related to the adsorption sites. The wavenumber of C=O vibration for gaseous CO is at around 2300 cm^{-1} , and once CO is adsorbed on the catalyst's surface, the wavenumber of C=O vibration is generally lower than 2200 cm^{-1} due to reduced bond order between C and O atom. The C-O vibration frequency in line site adsorption is ca. 2000 cm^{-1} ; the C-O vibration frequency in bridge site adsorption is ca. $1600 - 1900\text{ cm}^{-1}$ (**Figure R1-3d**). Moreover, the specific wavenumber of CO, adsorbed on catalyst's surface, depends heavily on the adsorption energy. **Generally speaking, the stronger the adsorption is, the weaker the C=O vibration, the lower its wavenumber, and the higher its vibration frequency.** The change in vibration frequency of C-O induced by CO adsorption on metals with different valence states has been studied (*Angew. Chem. Int. Ed.* 2021, 60, 15344, *J. Phys. Chem. C* 2019, 123, 5394, *Langmuir* 2004, 20, 10490). The *CO wavenumber on high spin CoPc is lower than that on low spin CoPc, which is in agreement with higher *CO vibrational frequency on high spin CoPc, meaning stronger CO adsorption.

Figure R1-3. a and b, Orbital interactions of CO molecule. c, Adsorption modes of CO on metal surface. d, The peak position assignments of different CO adsorption modes.

4. *The reviewer cannot get the meaning of spending such a great length of the manuscript to discuss the inverted C-H bonds on page 8. How did the authors assign the C-H peak? Is it possible that the C-H species comes from the organic contaminants since it seems irrelevant to the methanol production?*

Response: We thank the reviewer for raising these critical questions. Previous works have indicated that the inverted C-H peaks can be attributed to the consumption of adsorbed species, which is resulted from the subsequent reaction after their formation (*J. Am. Chem. Soc.* 2022, 144, 6613, *Chem* 2021, 7, 1297, *Angew. Chem. Int. Ed.* 2022, 61, e202206233). To rule out the possibility of C-H species coming from organic contaminants, electrolysis experiment was performed in H₂O and D₂O. The peaks at around 2950 cm⁻¹ showed a red-shift if H₂O was replaced by D₂O, the C-H bond disappeared at the same time and the C-D bond appeared, indicating that C-H species indeed came from CORR.

Figure R1-4. Operando ATR-SEIRAS spectra recorded over B-CoPc-400 in CO-saturated 0.5 M KHCO₃ (a) and KDCO₃ (b).

5. The high spin state of Co was achieved after thermal treatment. It seems that the temperature has a drastic impact. Did the authors compare the performance of catalysts treated with different temperatures?

Response: We thank the reviewer for the valuable suggestion. To figure out the effect of treatment temperature on CORR performance, the performance of catalysts treated at different temperatures (200 °C, 400 °C and 500 °C) were compared, among which, B-CoPc-400 delivered the best CORR performance (**Figure R1-5**). Experimental characterizations showed different spin states of Co in the catalysts treated at different temperatures. Together with DFT calculations, it is suggested that spin state of Co plays an important role in determining CORR performance.

Figure R1-5. Potential-dependent product selectivity for CO electroreduction catalyzed by B-CoPc-X (X represents treatment temperature).

6. Why did the authors use the nitrogen-doped carbon as the substrate? What if you use other typical carbon substrates such as Vulcan carbon black and graphene?

Response: We thank the reviewer for raising these questions. According to our previous research on CO₂ electroreduction, nitrogen-doped carbon worked well as a good catalyst's support to immobilize single atom catalysts. To study the effect of carbon support, typical carbon support of Vulcan carbon black was also examined. The experimental results indicate that the type of carbon supports had little influence on

CORR (Figure R1-6).

Figure R1-6. Potential-dependent product selectivity for CO electroreduction catalyzed by B-CoPc-400 and B-CoPc/C-400 (B-CoPc/C-400 represents Vulcan carbon black as support to prepare the catalyst).

7. On page 8, line 177, the authors claimed that the CORR over M-CoPc-400 was limited by proton transfer based on the KIE results and thought it was consistent with the operando ATR-SEIRAS results. However, the reviewer didn't see any corresponding results from SEIRAS study.

Response: We thank the reviewer for the valuable comment. *Operando* ATR-SEIRAS spectra showed that over M-CoPc-400, the peak intensity of *CO was strong, while the peak intensity of *CHO was weak, suggesting that the hydrogenation of *CO over M-CoPc-400 to *CHO was unfavorable. The conclusion deduced from SEIRAS study is consistent with the KIE experiment, suggesting that CORR was limited by proton transfer over M-CoPc-400.

8. The related NMR spectra of the resulting electrolytes after CO electrolysis and KIE experiments should be provided.

Response: We thank the reviewer for the valuable suggestion. According to the reviewer's suggestion, the resulting electrolytes after CO electrolysis were quantified by H-NMR and the results showed that methanol was the only liquid product (**Figure R1-8 a-c**). KIE experiment was also performed to examine the relation between kinetics of CORR and proton-feeding. Since KIE experiment mainly studies the effect of proton, deuterium water is used in the reaction process. However, the spin quantum number of

deuterium is 1, and the signal of protium element is measured by proton nuclear magnetic resonance spectroscopy, and deuterium will not show a peak. Thus, there is no way to provide its hydrogen spectrum. For the quantification in the KIE experiment, it is mainly by means of high-performance liquid chromatography (HPLC), and the experimental results are shown in **Figure R1-8d**.

Figure R1-8. a. Linear relationship between methanol concentration and relative area versus DMSO internal standard. ¹H-NMR spectrum for methanol determination of B-CoPc-400 (b) and M-CoPc-400. (c). HPLC spectra.

Reviewer #2 (Remarks to the Author):

In this manuscript, Ding et al. engineered the conformation of the cobalt active site within cobalt phthalocyanine (CoPc) molecular catalyst and achieved an improved performance for CO electroreduction to methanol. Specifically, the authors suggested that the catalytic activity of CoPc can be manipulated by tuning the spin-state of the Co 3d electrons. When the electron rearrangement of the Co 3d orbitals of CoPc change from the low-spin state to the high-spin state, the catalyst TOF was observed to increase by 5 times. Besides, the ATR-SEIRAS and DFT studies all supported this argument. Overall, I support the publication of this manuscript in Nature Communication provided the authors can address the following minor issues.

Response: We sincerely appreciate the reviewer for the time and efforts spent in assessing our work. To fully address the reviewer's concerns, detailed responses and supporting data are provided below.

Comments: 1. *While the M-CoPc and B-CoPc were purchased from chemical vendors, it is suggested to carry out some basic physical characterizations on them to confirm their chemical structures and purity, i.e., MS, NMR.*

Response: We thank the reviewer for the nice suggestion. We have carried out solid-NMR and MS measurements to confirm the structures and purity of M-CoPc and B-CoPc. For M-CoPc (**Figure R2-1a** and **b**): HRMS (EI) Calcd. for $C_{32}H_{16}CoN_8^+$ [M]⁺: 571.0830, found: 571.0844; ¹³C NMR (101 MHz) δ (ppm) = 171.4, 168.2, 123.3, 117.3, 115.1, 113.1, 99.9. For B-CoPc (**Figure R2-1c** and **e**): HRMS (EI) Calcd. for $C_{58}H_{27}Co_2N_{16}^+$ [M]⁺: 1065.1269, found: 1065.1277. ¹³C NMR (101 MHz) δ (ppm) = 174.3, 169.7, 137.7, 131.4, 125.3. The original NMR and MS spectra have been added to the revised Supplementary Information.

Figure R2-1. HR-MS (a, c) and ^{13}C NMR (b, d) spectra for M-CoPc and B-CoPc, respectively.

2. It seems that the authors have measured the Co-loading of M-Co-Pc and B-CoPc on nitrogen doped carbon, before and after the annealing, for estimating the TOF, however, these data are not available. Please provide these data and discuss whether the Co-loading plays a role in the catalysis.

Response: We thank the reviewer for the valuable comment and suggestion. We are very sorry for missing the data of Co loading and TOF in the manuscript. We calculated the TOF according to the calculation method given in the manuscript, and the results are shown in **Figure R2-2**, which reflects that the loading amount of Co plays a negligible role in affecting the electrochemical performance. Instead, the spin state of Co plays an important role in determining the electrochemical performance.

Figure R2-2. Potential-dependent product selectivity for CO electroreduction over B-CoPc-400 with different Co loading amount.

3. It is a bit surprise to see that methanol formation on B-CoPc-400 is limited by the desorption step. If it is the case, shouldn't the authors see more C-based species from ATR-SEIRAS, e.g., *CO, *C-H species, for the sample of B-CoPc-400, which is not the case. Besides, the authors attributed the "inverted C-H peaks" on B-CoPc-400 to the "consumption of adsorbed species as compared to the condition of OCP", this is hard to believe, one would not expect the formation of C-H species at OCP conditions. Moreover, as shown in Fig. S8, these "inverted C-H peaks" already exist in absence of CO in Ar-saturated 0.5 M KOH.

Response: We thank the reviewer for the valuable comments. All *operando* ATR-SEIRAS spectra were re-collected to ensure the measurement accuracy. As the reviewer stated, a few intermediates of CORR, e.g., *CO, *CHO and *C-H species, were observed over B-CoPc-400 from ATR-SEIRAS (**Figure R2-3 a-c**). Due to faster hydrogenation process over B-CoPc-400, less *CO and *CHO species were observed over B-CoPc-400 during CORR. Besides, previous works had indicated that inverted C-H peaks could be attributed to the consumption of adsorbed species (*Angew. Chem. Int. Ed.* 2022, 61, e202213296, *J. Am. Chem. Soc.* 2022, 144, 6613, *Chem* 2021, 7, 1297). Moreover, to further ensure whether there is a corresponding peak at 0 V (vs. RHE), we analyzed the potential at the open circuit voltage and the corresponding current at 0 V (vs. RHE). As shown in **Figure R2-3d** and **e**, at the open circuit voltage, the voltage is above 0 V (vs. RHE), and at 0 V (vs. RHE), there has already appeared a

reactive current, suggesting that at 0 V (vs. RHE), the electroreduction reaction of CO has already started. Thus, the corresponding signal peak can be detected by *operando* ATR-SEIRAS. **Figure R2-4** show the absence of *CO and *C-H peaks in Ar-saturated 0.5 M KOH.

Figure R2-3. *Operando* ATR-SEIRAS spectra recorded over B-CoPc-400 in CO-saturated 0.5 M KHCO₃ (a. H₂O, b. D₂O, and c. H₂O + D₂O). d, Open circuit potential-time curve for B-CoPc-400 and M-CoPc-400. e, The corresponding current-time curve at 0 V (vs. RHE).

Figure R2-4. *Operando* ATR-SEIRAS spectra recorded on M-CoPc-400 (a) and B-CoPc-400 (b) in Ar-saturated 0.5 M KHCO₃.

Reviewer #3 (Remarks to the Author):

*The study, entitled "Atomic High-Spin Co (II) Center for Highly Selective Electrochemical CO Reduction to CH₃OH," by Ding et al., reported a high-spin state binuclear cobalt phthalocyanine on nitrogen-doped carbon support (B-CoPc-400), which exhibits superior performance in the electroreduction of CO to CH₃OH, with a partial current density of 35 mA cm⁻² and a faradaic efficiency of 50% at -0.8 V vs RHE. Through the combination of Operando measurements and density functional theory calculations, the authors uncovered the mechanism by which the high-spin state of Co²⁺ weak the C-O bonding through electron accumulation on *CO, thereby improving its CORR performance. The paper is well-written and provides a comprehensive understanding of energy conversion through molecular catalysis. Thus, the manuscript is recommended for publication in Nature Communications after addressed the following questions.*

Response: We sincerely appreciate the reviewer for the time and efforts spent in assessing our work. To fully address the reviewer's concerns, detailed responses and supporting data are provided below.

1. In line 26, the term "RHE" (reversible hydrogen electrode) should be introduced and subsequently used as an abbreviation throughout the manuscript.

Response: We thank the reviewer for the helpful comment. The full name of "RHE" has been introduced when it appears for the first time in the revised manuscript.

2. In lines 51-52, the claim that "Cobalt phthalocyanine (CoPc) molecular catalyst showed the capability to electrochemically reduce CO to produce CH₃OH" is supported by references 19-21, which however pertain to the electrochemical reduction of CO₂. It is suggested that the author carefully check and replace these references with more appropriate ones that support the claims made.

Response: We thank the reviewer for the constructive comment. Accordingly, the references have been updated in the revised manuscript.

3. The in-situ Raman spectrums of M-CoPc-RT and B-CoPc-RT during thermal treatment, as presented in Fig. 1b and Fig. 1c, respectively, both show a weakened

intensity of the D and G band at 400 °C. An explanation of this observation should be provided by the authors.

Response: We thank the reviewer for the nice suggestion. It is true that the intensity of Raman signal became slightly weakened as treatment temperature increased. While it is also noticed that the ratio of the D to G band did not change dramatically with treatment temperature, suggesting that the structure of carbon support for M-CoPc and B-CoPc did not change dramatically. Therefore, decreasing Raman signal with increasing treatment temperature could be possibly due to change of macroscopic physical state (such as degree of aggregation) of the Raman sample and environmental fluctuation caused by temperature.

4. The FT-EXAFS spectra of B-CoPc-400 reveal new peaks at ~1.7 Å and ~2.2 Å in comparison to B-CoPc-RT, while M-CoPc-400 does not show significant differences from M-CoPc-RT. Further explanation is suggested.

Response: We thank the reviewer for the very constructive suggestion. The Co K-edge XANES spectrum of M-CoPc-RT/400 is similar to that of CoPc, indicating that the M-CoPc-RT/400 possesses the same Co oxidation state and D_{4h} symmetry as CoPc. However, the intensity of peak B (1s→4p_z transition), which is a fingerprint of square-planar metal-N₄ structure, is slightly weaker in B-CoPc-RT/400, suggesting a slightly distorted D_{4h} symmetry of the Co atom (*Angew. Chem. Int. Ed.* 2022, 61, e202213296, *Nat. Commun.* 2020, 11, 4233, *J. Am. Chem. Soc.* 2022, 144, 16131, *Inorg. Chem.* 1991, 30, 920). Furthermore, the B-Co-RT/400 displays a remarkably larger intensity ratio of peak C to peak D (I_C/I_D) than M-CoPc-400 (peak C and peak D can be assigned to the 1s→4p_{x,y} transitions and multiple scattering processes, respectively). Increased I_C/I_D had been shown to be beneficial towards promoting electrochemical reactions (*Nat. Commun.* 2021, 12, 4088, *J. Am. Chem. Soc.* 2021, 143, 11317, *Nat. Energy* 2018, 3, 140, *ACS Nano* 2015, 9, 12496). Moreover, as compared to M-CoPc, after thermal treatment at 400 °C, XANES and FT-EXAFS spectra for B-CoPc display a larger variation; B-CoPc-400 reveals new peaks at ~2.2 Å in comparison to B-CoPc-RT (as shown in **Figure R 3-1a** and **b**), indicating an increase of Co-N bond length,

highlighting the structure change driven by the thermal treatment.

Figure R3-1. a, Co K-edge XANES spectra before and after annealing. b, The corresponding Fourier transformation (FT)-EXAFS spectra.

5. The reference potentials in the manuscript should be presented consistently, either with or without brackets. (e.g., line 131 and line 134)

Response: We thank the reviewer for the suggestion. All reference potentials in the manuscript have been revised to “(vs. RHE)”.

6. A comparison of the performance with recently reported articles would be beneficial. Thus, a table summarizing this comparison could be included in the supplementary information.

Response: We thank the reviewer for the nice suggestion. According to the reviewer's suggestion, we made a comparison of CORR performance between our prepared catalyst and those reported in the literature (**Table R3-1**).

Table R3-1. Comparison of CORR performance between our prepared catalyst and those reported in the literature.

Catalyst	Current density (mA)	FE _{methanol} (%)	Reference
B-CoPc-400	154	57	This work
CoPc	4.77	14.3	Angew. Chem. Int. Ed. 2019, 58, 16172
CoPc/MWCNT	3.8	14	Chem. Eur. J. 2022, 28, e202200697

7. The system in which the LSV curves were measured, whether a flow cell or a typical three-electrode cell, should be specified. The reason for measuring the curves on the carbon paper instead of the GDE or glassy carbon rotating disc electrode should also be discussed.

Response: We thank the reviewer for the nice suggestion. CV and LSV measurements were performed on carbon paper to investigate the intrinsic catalytic activity of the as-constructed catalysts. The stability test was conducted on a flow cell, which was conducive to mass transport to meet the application requirement.

8. The authors should confirm whether the saturated calomel reference electrode was calibrated before the electrocatalytic performance test.

Response: We thank the reviewer for the comment. The saturated calomel reference electrode ($E_{RHE} = E_{SCE} + 0.0591 \times pH + 0.244$) was calibrated before the electrochemical measurement. The calibration data is shown in **Figure R3-2**.

Figure R3-2. Cyclic voltammograms of Pt electrode at a scan rate of 1 mV/s in H₂ saturated 0.1 M KCl solutions.

9. The total FE shown in Fig. 2c do not appear to be 100%. It would be useful to provide a table of graph showing all the product, including any other gas/liquid products that may have been generated.

Response: We thank the reviewer for the nice suggestion. **Table R3-2** shows the FE of all CORR products, including all gaseous and liquid products. The main reasons for the total FE lower than 100% include: (1) at low cathodic potentials, the current is small, and there could be large experimental errors; (2) some volatile liquid products (such as alcohols) could pass through the gas diffusion layer and be taken away by CO; (3) liquid products could also pass through ion exchange membrane and enter the anode compartment, which would not be collected and detected.

Table R3-2. FE of all the products.

Catalysts	Potential (V vs. RHE)	FE(H ₂)	FE(CH ₃ OH)	FE (total)
M-CoPc-400	-0.4	48%	5%	53%
	-0.5	74%	11%	85%
	-0.6	78%	13%	91%
	-0.7	75%	19%	94%
	-0.8	80%	16%	96%
	-0.9	92%	1%	93%
	-1.0	93%	2%	95%
B-CoPc-400	-0.4	35%	8%	43%
	-0.5	43%	44%	87%
	-0.6	33%	57%	90%
	-0.7	38%	56%	94%
	-0.8	44%	48%	92%
	-0.9	90%	4%	94%
	-1.0	92%	2%	94%

10. The authors are suggested to include ¹H NMR graphs for Fig. 2c.

Response: We thank the reviewer for the valuable suggestion. ¹H NMR graphs for Fig. 2c have been supplemented (**Figure R3-10**) to confirm the distribution and FE of products.

Figure R3-10. ^1H 1D NMR spectra of products and DMSO at different potentials (vs. RHE) for B-CoPc-400 (a) and M-CoPc-400 (b), respectively.

11. Have the authors measured the CORR performance of B-CoPc-RT? Otherwise, it is hard to say that high-spin state B-CoPc (B-CoPc-400) can catalyze CORR to methanol much more effectively than low-spin state B-CoPc (B-CoPc-RT).

Response: We thank the reviewer for the critical comment. The CORR performance of B-CoPc-RT was evaluated (**Figure R3-11**). According to the reviewer's comment, we examined the performance of CO electroreduction to CH_3OH over B-CoPc treated at different temperatures (RT, 200 °C, 400 °C, 500 °C) as shown in **Figure R3-11**. The experimental results show that as temperature increased, the selectivity of CH_3OH increased, but after exceeding 400 °C, the CH_3OH selectivity decreased significantly. The FE of CH_3OH over high-spin state B-CoPc (B-CoPc-400) was much higher than the counterpart low-spin state B-CoPc (B-CoPc-RT), confirming that high-spin state B-CoPc (B-CoPc-400) could catalyze CORR to methanol much more effectively than low-spin state B-CoPc (B-CoPc-RT).

Figure R3-11. Potential-dependent product selectivity for CO electroreduction catalyzed by B-CoPc-X (X represents treatment temperature).

12. *Characterization of the samples after the stability test, such as HRTEM and XAS, would be valuable to confirm whether the Co retains its atomic dispersion after the CORR.*

Response: We thank the reviewer for the helpful suggestion. HRTEM, HADDF-STEM and XAS have been performed on the catalyst after stability test. HRTEM and HADDF-STEM did not show appearance of nanoparticles (**Figure R3-12a and b**), indicating that after CORR, the catalyst remained atomic dispersion. Meanwhile, the near-edge features were basically maintained, and Co-Co peak could not be observed in the XAS spectra (**Figure R3-12d**), confirming that Co retained its atomic dispersion after CORR (**Figure R3-12c and d**). The maintenance of Co atomic dispersion after CORR corroborated the robust stability of B-CoPc-400.

Figure R3-12. HRTEM (a) and HADDF-STEM (b) of B-CoPc-400 after CORR. c. Co K-edge XANES spectra for B-CoPc-400 before and after CORR. d, The corresponding Fourier transformation (FT)-EXAFS spectra.

13. *To improve the comprehensiveness, this close literature should be cited: Chem Catalysis 2, 372-385 (2022); Angew. Chem. Int. Ed. 61, e202110186 (2022). Adv. Mater. 34, 2106541 (2022).*

Response: We thank the reviewer for the suggestion. These references have been cited correspondingly in the revised manuscript.

14. *It appears that some of the figure numbers in the manuscript may be incorrect. For example, in line 202, it should be Fig. 4a instead of Fig. 3a. The same applies to lines 203, 205, 211, and 219. The figure number in line 213 is missing. The authors are advised to carefully check all the figure numbers.*

Response: We thank the reviewer for the valuable comment. We have carefully

checked the entire manuscript and made the corresponding corrections.

15. The style of describing figures in the manuscript should be standardized, either as Fig. xx or Figure xx.

Response: We thank the reviewer for the valuable comment. The style of describing figures in the manuscript has been standardized as “Figure”.

16. In line 343, the authors state that all LSV data were corrected for the ohmic drop. The accurate value of the ohmic drop, as well as the method used to measure it, should be reported.

Response: We thank the reviewer for the valuable comment. The Ohmic resistance was measured by electrochemical impedance spectroscopy (EIS) at open circuit voltage. In the EIS measurement, a sinusoidal AC perturbation of 5 mV was applied to the electrode over the frequency range from 0.5 to 105 Hz. The typical Ohmic resistance of the electrode is about 12.5 Ω in CO saturated 0.5 M bicarbonate solution. Before each impedance measurement, the workstation program automatically measured the open circuit voltage of the system and recorded the impedance spectrum at the open circuit voltage (**Figure R3-13**). Ohmic drop was compensated using the method described in literature. A square wave was applied to the potentiostat from the function waveform generator, then the potentiostat modulated this signal and added a correction to the applied voltage according to the current response.

Figure R3-13. The EIS spectrum for M-CoPc-RT/400 and B-CoPc-RT/400.

Reviewer #4 (Remarks to the Author):

Recommendation: Publish after major revisions noted.

*In this study, the author designed the structure of Co active center in cobalt phthalocyanine (CoPc) molecular catalyst and obtained the B-CoPc-RT/400 catalyst with significantly optimized catalytic performance of electrochemical carbon monoxide (CO) reduction to methanol. Through a series of experimental characterization and density functional theory (DFT) calculations, it is shown that the Co 3d orbitals of CoPc transferred from the low-spin state (LS, 1/2) to the high-spin state (HS, 3/2), induced by the electron rearrangement Co for the change of molecular conformation, which weakens the C-O bond in *CO, promotes the hydrogenation of the CORR intermediate, and is responsible for greatly enhancing the CORR performance. However, this study has methodological problems and limited innovation, thus the paper is recommended to be major revised. The detailed comments are as follows:*

Response: We sincerely appreciate the reviewer for the time and efforts spent in assessing our work. To fully address the reviewer's concerns, detailed responses and supporting data are provided below.

1. The study constructed cobalt phthalocyanine and binuclear cobalt phthalocyanine (M-CoPc and B-CoPc) catalyst anchored on nitrogen doped carbon support, while the reason to choose the binuclear molecule and monoatomic Co catalyst to study is not mentioned, which can be added in the introduction.

Response: We thank the reviewer for the valuable suggestion. "The spatial structure of binuclear cobalt phthalocyanine is very different from that of mononuclear cobalt phthalocyanine, which is anticipated to result in distinct crystal field and thus different catalytic performance, making binuclear cobalt phthalocyanine an interesting candidate to be investigated for CORR." has been added into the introduction in the revised manuscript.

2. In DFT calculation, the authors believed that the spin state transition was caused by geometric distortion of CoPc. However, there is no any experimental evidences that support the structural distortion of CoPc.

Response: We thank the reviewer for the valuable comment. In this work, to understand the chemical states and distortion of CoPc in M-CoPc-RT/400 and B-CoPc-RT/400, X-ray absorption near edge structure (XANES), extended X-ray absorption fine structure (EXAFS) and L-edge characterizations were performed. As shown in **Figure R4-1a** (**Figure 1g and f in the manuscript**). The Co K-edge XANES spectrum of M-CoPc-RT/400 is similar to that of CoPc, indicating that the M-CoPc-RT/400 possesses the same Co oxidation state and D4h symmetry as CoPc. However, the intensity of peak B ($1s \rightarrow 4p_z$ transition) that is a fingerprint of square-planar metal-N₄ structure is slightly weaker in B-CoPc-RT/400, suggesting a slightly distorted D4h symmetry of the Co atom (*Angew. Chem. Int. Ed.* 2022, 61, e202213296, *Nat. Commun.* 2020, 11, 4233, *J. Am. Chem. Soc.* 2022, 144, 16131, *Inorg. Chem.* 1991, 30, 920). Meanwhile, **Figure R4-1b** displays the Fourier transformed EXAFS spectra. The peak at 1.55 Å for M-CoPc-RT/400 is the scattering of the first Co-N shell. Notably, a new peak (at 2.2 Å) was observed in the Fourier transformed EXAFS spectra, indicating breaking of D4h symmetry for B-CoPc-400, which was caused by molecular distortion induced by heat treatment (*Nat. Commun.* 2021, 12, 4088, *J. Am. Chem. Soc.* 2021, 143, 11317, *Nat. Energy* 2018, 3, 140, *ACS Nano* 2015, 9, 12496). As shown in **Figure R4-1c**, L_{2,3}-edge XAS spectra were further collected to study the structure change of M-CoPc and B-CoPc after 400 °C treatment. Three features appear in the energy region of Co L₃-edge, denoted as A₁, A₂ and A₃ as shown in **Figure R4-1c and d**. The peak A₁ can be assigned to the transition from 2p_{3/2} to the 3d_{z²} orbitals, while the peaks A₂ and A₃ are originated from the transition to 3d_{x²-y²} orbitals (*J. Chem. Phys.* 2012, 137, 054306). The Co L_{2,3}-edge XAS spectrum of M-CoPc-RT and M-CoPc-400 are similar, which display the same characteristic feature as that of CoPc (*J. Phys. Chem. C.* 2015, 119, 27569, *J. Phys. Chem. C.* 2017, 121, 26372).

Figure R4-1. a, Co K-edge XANES spectra before and after annealing. b, The corresponding Fourier transformation of (FT)-EXAFS spectra. (c, d) The experimental and simulated cobalt $L_{2,3}$ -edge XAS spectra of M-CoPc-RT/400 and B-CoPc-RT/400, respectively. (e, f) Schematic illustration showing the effect of thermal treatment on the coordination environment of M-CoPc-RT/400 and B-CoPc-RT/400, respectively.

3. From the results of the spin state change in the paper, it can be seen that the spin state of Co^{2+} changes from L.S. to H.S. after the transformation of B-CoPc-RT into B-CoPc-400 after heat treatment, while the spin state of M-CoPc-RT does not change significantly before and after heat treatment. Please clarify that the structure distortion of binuclear molecules and the heat treatment are the driving force of the spin state change. (*Journal of Materials Chemistry A* 2019, 7 (47), 27175-27185; *Journal of the American Chemical Society* 2018, 140 (45), 15149-15152; *Energy & Environmental Science* 2022, 15 (4), 1601-1610; *Nature Communications* 2021, 12 (1), 1734).

Response: We thank the reviewer for the valuable comment and suggestion. We have carefully studied the valuable references provided by the reviewer as well as the literature. It suggests that annealing can induce changes in the molecular structure of phthalocyanines and changes in coordination bonds. For example, in the study

performed by Wang et al., when the nanowires formed by NiPc were heated at about 400 °C, although the crystal structure of NiPc remained basically unchanged, its ultraviolet absorption characteristics changed, which might be due to the distortion induced by thermal annealing (*Mater. Res.*, 2016, 3, 125002). Miller et al. studied FePc that was deposited onto Ketjen black carbon and heated for 2 hours in inert atmosphere (Ar) at different temperatures (400, 500, 600, 700, 800, 900 and 1000 °C) (*Phys. Chem. Chem. Phys.* 2016, 18, 33142). The atomic structure of Fe in each sample was determined by XAS and correlated to the activity of ORR and ORR mechanisms. The results show that the samples prepared at 600 and 700 °C have the highest electrochemical catalytic activity for ORR, consistent with the finding that the FeN₄ active sites are thermally stable up to 700 °C. More importantly, when FePc was treated at 400 °C, its XAS change characteristics were basically consistent with CoPc in our experiments. Thus, based on both our experimental characterizations and early studies in the literature, it is concluded that the structure distortion of binuclear molecule induced by heat treatment changes its spin state.

4. Please give other characterization verification of binuclear molecule structure of B-CoPc-RT/400 in addition to HAADF-STEM images.

Response: We thank the reviewer for the nice suggestion. According to the reviewer's suggestion, we have carried out solid-NMR and MS measurements to confirm the structures and purity of M-CoPc and B-CoPc. For M-CoPc (**Figure R4-4a** and **b**): HRMS (EI) Calcd. for C₃₂H₁₆CoN₈⁺ [M]⁺: 571.0830, found: 571.0844; ¹³C NMR (101 MHz) δ (ppm) = 171.4, 168.2, 123.3, 117.3, 115.1, 113.1, 99.9. For B-CoPc (**Figure R4-4c** and **c**): HRMS (EI) Calcd. for C₅₈H₂₇Co₂N₁₆⁺ [M]⁺: 1065.1269, found: 1065.1277. ¹³C NMR (101 MHz) δ (ppm) = 174.3, 169.7, 137.7, 131.4, 125.3. The original NMR and MS spectra have been added to the revised Supplementary Information.

Figure R4-4. HR-MS (a, c) and ^{13}C NMR (b, d) spectra for M-CoPc and B-CoPc, respectively.

5. *The transition of spin state of Co^{2+} should be also characterized by other characterization techniques to measure the magnetic property of the catalysts, such as the magnetic susceptibility.*

Response: We thank the reviewer for the valuable suggestion. To confirm the spin state transition of Co^{2+} , electron paramagnetic resonance (EPR) measurements were further conducted to obtain information on the unpaired electron of Co ions in the catalyst (**Figure R4-5**). The EPR spectrum of M-CoPc-RT and B-CoPc-RT show a strong signal at $g = 2.71$, which is a typical characteristic signal of low-spin Co^{2+} , suggesting that unpaired electron exists in M-CoPc-RT and M-CoPc-RT. In addition, this signal still exists when M-CoPc-RT is annealed at $400\text{ }^\circ\text{C}$ to obtain M-CoPc-400, indicating that Co in M-CoPc-400 is still at the low-spin state. However, when B-CoPc-RT is annealed at $400\text{ }^\circ\text{C}$ to obtain B-CoPc-400, the signal at $g = 2.71$ reduces, and at the same time, a new signal appears at $g = 2.00$, which is a typical high-spin Co^{2+} signal. The EPR results suggest that the $400\text{ }^\circ\text{C}$ treatment of B-CoPc-RT changes its spin structure from low spin to high spin, in agreement with $\text{L}_{2,3}$ -edge analysis.

Figure R4-5. EPR spectra of various samples.

6. In Table S1, the CN of Co-N in B-CoPc-400 is significantly smaller than B-CoPc-RT, and even M-CoPc-400, which is irrational, please give the explanation.

Response: We thank the reviewer for the valuable comment. We are very sorry for the confusion caused by the unclear table specification that we made. As shown in **Table R4-1**, the CN of Co-N in B-CoPc-400 is 4.7 (4.1 + 0.6), and the CN of Co-N in B-CoPc-400 is 4.7.

Table R4-1. Structural parameters extracted from the Co K-edge EXAFS fitting. ($S_0^2 = 0.76$).

Sample	Scattering pair	CN	R(\AA)	$\sigma^2(10^{-3}\text{\AA}^2)$	$\Delta E_0(\text{eV})$	R factor (10^{-2})
M-CoPc-RT	Co-N	4.2	1.89	3.0	3.4	6.7
M-CoPc-400	Co-N	4.7	1.87	3.0	-6.2	5.4
B-CoPc-RT	Co-N	4.7	1.92	8.0	-8.1	1.8
B-CoPc-400	Co-N	4.1	2.10	5.4	1.4	1.6
	Co-N	0.6	1.90	5.4	1.4	1.6

For the EXAFS fitting in Table R4-1, S_0^2 is the amplitude reduction factor; CN is the coordination number; R is interatomic distance; σ^2 is Debye-Waller factor; ΔE_0 is edge-energy shift. R factor is used to evaluate the goodness of fitting.

S_0^2 : This value was fixed during EXAFS fitting, based on the known structure of Co foil. Error bars that characterize the structural parameters obtained by EXAFS were estimated as $N \pm 20\%$; $R \pm 1\%$; $\sigma^2 \pm 20\%$; and $\Delta E_0 \pm 20\%$.

7. In Fig 2 and Fig 3, the comparison of catalytic performance and the discussion of CORR mechanism are mainly for M-CoPc-400 and B-CoPc-400, while Fig. 4 points out that the spin-state transition of B-CoPc occurs before and after heat treatment. Please clarify the rationality of the performance comparison sample.

Response: We thank the reviewer for the valuable comment. The CORR performance of B-CoPc-RT was also measured. The result showed that FE of methanol over B-CoPc-RT was much lower than its counterpart B-CoPc-400, suggesting that the high-spin B-CoPc was more effective than low-spin B-CoPc in catalyzing CORR to methanol (**Figure R4-6**).

Figure R4-6. Potential-dependent product selectivity for CO electroreduction catalyzed by B-CoPc-X (X represents treatment temperature).

8. Please explain in detail how to obtain KIE values of different catalysts and the meaning of these values.

Response: We thank the reviewer for the valuable comment. To examine the relation between kinetics of CORR and proton-feeding, the kinetic isotope effect (KIE) of H/D was measured using H_2O and D_2O as the proton sources, which was performed at -0.6

V (vs. RHE). When D₂O was used to replace H₂O in 0.5 M KOH or 0.5 M K₂SO₄ electrolyte, the partial current density of methanol was significantly reduced over M-CoPc-400. The KIE value refers to the ratio of the methanol partial current density recorded in H₂O to that recorded in D₂O. As shown in Figure 3f, compared to B-CoPc-400, M-CoPc 400 displays a much higher KIE value of 2.10 and 2.13 in KOH and K₂SO₄ solution, respectively, indicating that the CORR over M-CoPc-400 was limited by proton transfer.

*9. In line 153-155, the author claimed that “The *CO vibrational frequency on M-CoPc-400 is slightly lower than that on B-CoPc-400 (1911 and 1951 cm⁻¹ at 0 V vs. RHE), indicating a slightly higher binding strength of B-CoPc-400 towards CO.”. May the authors provide evidence or literature reports to support this conclusion. Furthermore, in line 258-260, it was proposed by DFT study that “As compared to LS-Co²⁺, 3d_{z²} orbital of HS-Co²⁺ is fully occupied, resulting in weakened CO adsorption over HS-Co²⁺, consistent with the DFT calculation.” Is that contradicting to the results of operando ATR-SEIRAS measurement?*

Response: We thank the reviewer for raising the critical comment. According to the molecular orbital theory, the highest occupied molecular orbital (HOMO) of CO is the almost non-bonding σ orbital that is localized to C, and the lowest unoccupied molecular orbital (LUMO) is the antibonding π orbital. The involvement in formation of σ and π orbital are C and O atoms. **Figure R4-7a** displays the molecular orbital energy level of CO molecule, and the ground state configuration is: $1\sigma^2 2\sigma^2 1\pi^4 3\sigma^2$. The 1σ orbital is primarily localized on O, thus, they are essentially non-bonding or weakly bonding orbitals. 2σ orbital is bonding orbital. The 1π orbital is a double degenerate pair of π bonding orbitals, which mainly has the properties of C_{2p} orbital. The HOMO of CO is a 3σ orbital, which has the properties of a C_{2pz} orbital, basically a non-bonding orbital, and is localized to C. The LUMO is a double degenerate pair of antibonding π orbitals, which has the properties of a C_{2p} orbital (**Figure R4-7b**). This combination of frontier orbitals (a pair of empty π orbitals that are essentially localized to carbon's fully filled σ orbitals) is the reason to form M_n-CO. CO is a ligand with a variety of

coordination modes (**Figure R4-7c**), usually involving bridging one, two, or three metal atoms, and the expansion frequency of CO follows the following sequence: $MCO > M_2CO > M_3CO$. The coordination information shows that CO has excellent adsorption on the surface of catalysts. From previous studies, it can be seen that the CO adsorption energy and the C-O vibration frequency are related to the adsorption sites. The wavenumber of C=O vibration for gaseous CO is at around 2300 cm^{-1} , and once CO is adsorbed on the catalyst's surface, the wavenumber of C=O vibration is generally lower than 2200 cm^{-1} due to reduced bond order between C and O atom. The C-O vibration frequency in line site adsorption is ca. 2000 cm^{-1} ; the C-O vibration frequency in bridge site adsorption is ca. $1600 - 1900\text{ cm}^{-1}$ (**Figure R4-7d**). Moreover, the specific wavenumber of CO, adsorbed on catalyst's surface, depends heavily on the adsorption energy. **Generally speaking, the stronger the adsorption is, the weaker the C=O vibration, the lower its wavenumber, and the higher its vibration frequency.** The change in vibration frequency of C-O induced by CO adsorption on metals with different valence states has been studied (*Angew. Chem. Int. Ed.* 2021, 60, 15344, *J. Phys. Chem. C* 2019, 123, 5394, *Langmuir* 2004, 20, 10490). The *CO wavenumber on high spin CoPc is lower than that on low spin CoPc, which is in agreement with higher *CO vibrational frequency on high spin CoPc, meaning stronger CO adsorption (**Figure R4-8**).

Figure R4-7. a and b, Orbital interactions of CO molecular. c, adsorption modes of CO molecules on metal surfaces. d, The peak position assignments of different CO adsorption modes on the infrared spectrum.

Figure R4-8. Operando ATR-SEIRAS spectra of CO reduction over (a) B-CoPc-400 and (b) M-CoPc-400 in CO-saturated 0.5 M KOH.

10. In line 210-216, “The theoretical spectra (Fig. 3d) were also calculated with the parameters ($10Dq = 1.3$ eV, $Ds = 0.36$ eV, $Dt = 0.10$ eV, $\Delta = 6.0$ eV for B-CoPc-RT and $10Dq = 0.1$ eV, $Ds = 0.09$ eV, $Dt = -0.01$ eV, $\Delta = 6.0$ eV for M-CoPc-400), and the results (Fig. d and Fig. S11c, d) reveal that the Co in B-CoPc-RT was still at the ground state in $2A1g$ symmetry ($dx^2-y^2dz^21dxy2dxz,yz4$), however, with severe distortion of coordination environment around Co after heat treatment, the $10Dq$ and Δ_{eg} reduced from 1.3 eV and 1.85 eV for M-CoPc-RT to 0.1 eV and 0.3 eV for B-CoPc-RT (Fig. S12b).”

1) Please confirm: is M-CoPc-400 here correct? Because there is no spectra for M-CoPc-400 in Figure 3d.

Response: We thank the reviewer for the comment. We are very sorry for the typo. Actually, we wanted to express the parameters for B-CoPc-RT and B-CoPc-400 in lines 210-212 and wrote B-CoPc-400 incorrectly as M-CoPc-400. We have made a careful check and changed the typo accordingly.

2) Please confirm: is M-CoPc-RT here correct? $10Dq = 1.3$ eV for B-CoPc-RT. $10Dq = 1.3$ eV for B-CoPc-RT.

Response: We thank the reviewer for raising the question. We have corrected our expression as “however, with distortion of coordination environment around Co after heat treatment, the $10Dq$ and Δ_{eg} reduced from 1.3 eV and 1.85 eV for B-CoPc-RT to 0.1 eV and 0.3 eV for B-CoPc-400” in our revised manuscript. We apologize for the

typo again.

3) Generally, $10Dq$ is called as crystal field splitting energy. Why the $10Dq$ is only 0.1 eV for B-CoPc-400 (if correct)?

Response: We thank the reviewer for raising the question. As stated by the reviewer, $10Dq$ is often used to describe the crystal field splitting energy, which reflects the energy difference between e_g orbitals and t_{2g} orbitals ($10Dq = E(e_g) - E(t_{2g})$). This energy difference is related to the symmetry of the crystal field. In our work, Co ion locates at the site with D_{4h} symmetry for M-CoPc-RT, and $10Dq$ can be expressed as $E(d_{x^2-y^2}) - E(d_{xy})$. In this case, the strong interaction between Co 3d orbital and the coordinated N 2p orbital mainly locates along the x/y axis, which will significantly raise the energy of $d_{x^2-y^2}$ orbital and has little effect on d_{xy} orbital, inducing a large $10Dq$ (2.5 eV). With coordinated N atoms deviating from the xy plane, the effect on $d_{x^2-y^2}$ orbital is gradually weakened. For B-CoPc-400, the energy difference between $d_{x^2-y^2}$ orbital and d_{xy} orbital is significantly reduced due to the deviation of N atoms from xy plane (**Figure R4-9**).

Figure R4-9. Energy of the cobalt center in different 3d electron configurations.

11. Please pay attention to the one-to-one correspondence between the caption and the figure in the text and the accuracy of language expression, for example, Fig 3d

appeared twice in the caption.

Response: We thank the reviewer for the valuable comment. The one-to-one correspondence between the caption and the figure as well as the accuracy of language expression have been carefully checked and revised correspondingly.

REVIEWER COMMENTS

Reviewer #1 (Remarks to the Author):

Q3 and Q4 are still not addressed.

In Q3, in general, the wavenumber is proportional to the vibration frequency. Therefore, it is necessary for the authors to reconsider the following statement: "Generally speaking, the stronger the adsorption is, the weaker the C=O vibration, the lower its wavenumber, and the higher its vibration frequency." Meanwhile, the explanation provided here by the authors still contradicts the IR results. Specifically, on page 8, the sentence "The *CO vibrational frequency on M-CoPc-400 is slightly lower than that on B-CoPc-400 (1911 and 1951 cm⁻¹ at 0 V vs. RHE), indicating a slightly higher binding strength of B-CoPc-400 towards CO" remains incorrect. A lower wavenumber or vibrational frequency of CO on M-CoPc-400 actually means a higher binding strength. So the CO binding strength on M-CoPc-400 (low spin CoPc) is greater than that on B-CoPc-400 (high spin CoPc).

In Q4, the authors did not present a compelling assignment for the C-H bond. The reviewer did not find any discussions regarding C-H peak assignment in the recommended references (J. Am. Chem. Soc. 2022, 144, 6613, Chem 2021, 7, 1297, Angew. Chem. Int. Ed. 2022, 61, e202206233).

Reviewer #2 (Remarks to the Author):

The reviewer appreciate the efforts made by the authors, most of the concerns have been addressed. The manuscript can be published.

One minor suggestion: can the authors try to estimate the thermodynamic potential for reducing CO to methanol? It is quite strange that the authors claim that the CO reduction occurs already at 0 V vs. RHE.

Reviewer #3 (Remarks to the Author):

The authors have well addressed the questions raised by the additional experiments and explanations. The quality of this work has been significantly improved. Thus, I recommend it publish in Nature Communications without further modification.

Reviewer #4 (Remarks to the Author):

The author has provided detailed responses to most of the questions, but there are still some issues in this manuscript that need improvement to meet the strict standards of Nature Communications. Some additional comments are as follows.

1. What is the basis for the degree of distortion (30°/15°) applied by the HS-Co model in the DFT calculation? The experiment only proves the distortion, but does not give the specific distortion degree.
2. The author attributes the enhanced activity to the enhanced CO adsorption of high spin Co sites, but the conclusion in DFT is that the adsorption of HS Co on CO is weakened (line 288). However, due to the anti-bonding feature of $3dxz/dyz-2\pi^*$, the C-O bond is weakened, which is beneficial for CORR. Is there a contradiction between the experiment and the DFT section? How to explain?
3. The expression in line 246 is incorrect: "suggesting that unpaired electron exists in M-CoPc-RT and M-CoPc-RT."

Reply to Reviewers

Reviewer #1 (Remarks to the Author):

Q3 and Q4 are still not addressed.

Response: We sincerely appreciate the reviewer for the time and efforts spent in evaluating our manuscript.

*In Q3, in general, the wavenumber is proportional to the vibration frequency. Therefore, it is necessary for the authors to reconsider the following statement: “Generally speaking, the stronger the adsorption is, the weaker the C=O vibration, the lower its wavenumber, and the higher its vibration frequency.” Meanwhile, the explanation provided here by the authors still contradicts the IR results. Specifically, on page 8, the sentence “The *CO vibrational frequency on M-CoPc-400 is slightly lower than that on B-CoPc-400 (1911 and 1951 cm⁻¹ at 0 V vs. RHE), indicating a slightly higher binding strength of B-CoPc-400 towards CO” remains incorrect. A lower wavenumber or vibrational frequency of CO on M-CoPc-400 actually means a higher binding strength. So the CO binding strength on M-CoPc-400 (low spin CoPc) is greater than that on B-CoPc-400 (high spin CoPc).*

Response: We thank the reviewer for the valuable comments. A lower wavenumber of C-O vibration in ATR-SEIRAS spectrum indicates a lower banding energy of C-O bond, meaning a higher binding strength of *CO with the catalytic center. The *operando* ATR-SEIRAS measurements showed that the *CO vibrational frequency was 1911 cm⁻¹ at 0 V vs. RHE on M-CoPc-400 (LS Co²⁺), and the *CO vibrational frequency was 1951 cm⁻¹ at 0 V vs. RHE on B-CoPc-400 (HS Co²⁺). The *CO vibrational frequency for B-CoPc-400 (HS Co²⁺) is higher than that for M-CoPc-400 (LS Co²⁺), indicating a lower binding strength of B-CoPc-400 to CO; this conclusion is consistent with the DFT calculation (Figure R1).

We are sorry for the typo in “The *CO vibrational frequency on M-CoPc-400 is slightly lower than that on B-CoPc-400 (1911 and 1951 cm⁻¹ at 0 V vs. RHE), indicating a slightly higher binding strength of B-CoPc-400 towards CO”. The description has

been revised as “The *CO vibrational frequency on M-CoPc-400 is slightly lower than that on B-CoPc-400 (1911 and 1951 cm⁻¹ at 0 V vs. RHE), indicating a slightly lower binding strength of B-CoPc-400 towards CO”. Overall, the DFT and *operando* ATR-SEIRAS results are consistent, and both suggest the binding strength of *CO on B-CoPc-400 (HS Co²⁺) is lower than that on M-CoPc-400 (LS Co²⁺). Moreover, this conclusion is also supported by the interactions between molecular frontier orbitals interactions between 5σ and 2π* of CO and the 3d orbitals of LS-Co²⁺ and HS-Co²⁺ (line 286-293).

In Q4, the authors did not present a compelling assignment for the C-H bond. The reviewer did not find any discussions regarding C-H peak assignment in the recommended references (J. Am. Chem. Soc. 2022, 144, 6613, Chem 2021, 7, 1297, Angew. Chem. Int. Ed. 2022, 61, e202206233).

Response: We thank the reviewer for the valuable comments. The C-H peak assignment can be found in *J. Am. Chem. Soc.* 2022, 144, 6613 (**Figure R1-1**). The wavenumber for the C-H bond is located between 2800 cm⁻¹ and 3000 cm⁻¹. The similar conclusion can also be found in *Langmuir* 1993, 9, 263-267 and *Adv. Phys. Chem.* 2012, 903272 (**Figure R1-2 and R1-3**).

Figure R1-1. Dynamical configuration of surfactants at the electrified electrode–electrolyte interface. (a) CV curves after chronoamperometry technique testing in KHCO_3 electrolytes with and without CTAB at a scan rate of 50 mV s^{-1} , in which the CTAB-containing system exhibited pseudocapacitance characteristics. (b) LSV curves of 0.5 M KHCO_3 electrolytes with and without 1 mM CTAB from -0.2 to -1.3 V versus RHE at a scan rate of 5 mV s^{-1} . (c) N 1s XPS spectra of the CTAB-containing system before and after being biased. (d) *In situ* ATR-SEIRAS spectra under various potentials for the CTAB-containing system in the range of $3000\text{--}2800 \text{ cm}^{-1}$. (e) Schematic illustration of surfactant configuration at the electrified interface. Random distribution at low potentials (top) and nearly ordered assembly at high potentials (bottom). Adopted from *J. Am. Chem. Soc.* 2022, 144, 6613.

Figure R1-2. FTIR spectra of CTAB on a self-supporting silica film at low surface coverage for (A) wet surface, (B) dry surface, and (C) rewet surface. Adopted from *Langmuir* 1993, 9, 263.

Figure R1-3. Selected regions of the FTIR spectra for the SDS molecule: (a) C–H stretching vibrational features; (b) methylene scissoring vibrational mode; (c) and (d) sulfate asymmetric and symmetric stretching bands, respectively. The spectra refer to the SDS packing densities: 1.2×10^{15} ; 4.8×10^{15} ; 3.5×10^{16} ; 4.3×10^{17} molecules cm^{-2} . Adopted from *Adv. Phys. Chem.* 2012, 903272.

Reviewer #2 (Remarks to the Author):

The reviewer appreciates the efforts made by the authors, most of the concerns have been addressed. The manuscript can be published.

Response: We sincerely appreciate the reviewer for the time and efforts spent in evaluating our manuscript.

One minor suggestion: can the authors try to estimate the thermodynamic potential for reducing CO to methanol? It is quite strange that the authors claim that the CO reduction occurs already at 0 V vs. RHE.

Response: We thank the reviewer for raising this valuable question. The methanol synthesis reactions are exothermic (*J. CO₂ Util.*, 2015, 10, 95-104), and the thermodynamic potential for reducing CO to methanol can be calculated based on the thermodynamic potential of CO₂ to CO and CO₂ to methanol (*Nat. Rev. Chem.* 2021, 5, 564–579) by the Gate's law:

The calculated thermodynamic potential for reducing CO to methanol is about 0.15 V (vs. RHE).

Reviewer #3 (Remarks to the Author):

The authors have well addressed the questions raised by the additional experiments and explanations. The quality of this work has been significantly improved. Thus, I recommend it publish in Nature Communications without further modification.

Response: We sincerely appreciate the reviewer for the time and efforts spent in evaluating our manuscript.

Reviewer #4 (Remarks to the Author):

The author has provided detailed responses to most of the questions, but there are still some issues in this manuscript that need improvement to meet the strict standards of Nature Communications. Some additional comments are as follows.

Response: We sincerely appreciate the reviewer for the time and efforts spent in evaluating our manuscript.

1. What is the basis for the degree of distortion (30°/15°) applied by the HS-Co model in the DFT calculation? The experiment only proves the distortion but does not give the specific distortion degree.

Response: We thank the reviewer for raising the valuable question. The distortion of CoPc was revealed by the Co K and L_{2,3}-edge XAS, while, the current characterization techniques cannot give the precise distortion degree, and thus theoretical calculation is used to simulate the effect of distortion degree on the spin state. For understanding the distortion of CoPc on the spin state, 0 to 30° of distortion were calculated, as shown in **Figure R4-1**. If no distortion is applied on CoPc, CoPc is in low spin state, which is almost impossible to tune to a high spin state because of the huge difference in free energy between these two states. In this case, the rate determining step of CO to methanol reduction is hydrogenation of *OCH₂ to form *OCH₃ and the corresponding free energy change is about 0.75 eV. When distortion (0 to 30°) is applied on CoPc, the energies of CoPc between high spin and low spin states become smaller. As the degree of distortion reaches and exceeds 15°, DFT calculation results show that the two states of HS and LS have the same energy, suggesting that the Co²⁺ center in CoPc can be easily transformed between HS and LS states, thus it would show two possible mechanisms of CORR (**Figure R4-2**). In this work, we take the CoPc with 15° of distortion as the example, there are two possible mechanisms, (LS-Co²⁺-(15°) and HS-Co²⁺-(15°)) for CORR to methanol. It was found that the high spin state could decrease the free energy change of rate determining step from 0.75 eV of *OCH₂ → *OCH₃ to 0.43 eV of the desorption of *CH₃OH. While the low spin state of CoPc with 15° of

distortion is mediocre as that of 0° of distortion. The $*\text{OCH}_2 \rightarrow *\text{OCH}_3$ needs 0.50 eV free energy change, and the rate determining step of LS-Co²⁺-(15°) is the desorption of $*\text{CH}_3\text{OH}$ corresponding to ΔG of 0.96 eV. In addition, the $*\text{CO} \rightarrow *\text{CHO}$ process needs 0.29 eV to overcome which is higher than that of high spin state of CoPc with 15° of distortion. To better understand the spin state of Co²⁺ center on the CORR over CoPc, we have added the reaction free energy curve of low spin state CoPc of distortion of 15° (LS-Co²⁺-(15°)) in **Figure R4-2**.

Figure R4-1. The effect of structure distortion on spin state of Co²⁺ in CoPc.

Figure R4-2. Potential energy profiles of CORR over LS-Co²⁺-(0 and 15°) and HS-Co²⁺-(15°).

2. The author attributes the enhanced activity to the enhanced CO adsorption of high spin Co sites, but the conclusion in DFT is that the adsorption of CO on HS Co is

weakened (line 288). However, due to the anti-bonding feature of $3d_{xz}/d_{yz}-2\pi^*$, the C-O bond is weakened, which is beneficial for CORR. Is there a contradiction between the experiment and the DFT section? How to explain?

Response: We thank the reviewer for the valuable comments. We are sorry that the intrinsic mechanism of the enhanced catalytic performance was not clearly described in the previous version of the article. In our work, both *operando* ATR-SEIRAS measurements and DFT calculations suggest that the binding strength of *CO on B-CoPc-400 (HS Co²⁺) is lower than that on M-CoPc-400 (LS Co²⁺). Moreover, this conclusion is also supported by the interactions between molecular frontier orbitals interactions between 5σ and $2\pi^*$ of CO and the 3d orbitals of LS-Co²⁺ and HS-Co²⁺ (line 286-293). We did not ascribe the catalytic activity enhancement to the enhanced CO adsorption. In this work, both KIE and DFT results indicate that the hydrogenation of CORR intermediates is energetically favorable over B-CoPc-400 (H-Co²⁺). Based on the charge density difference calculations and molecular frontier orbitals interactions analysis for *CO-LS-Co²⁺ and *CO-HS-Co²⁺, the weaker $3d_z^2-5\sigma$ and more electron transfer from Co to *CO via π back-donation (via $d_{xz}/d_{yz}-2\pi^*$ bond) for HS-Co²⁺ lead to smaller n_{CT} (net charge transfer) from CO to Co site, enabling more electron accumulation on the $2\pi^*$ orbital of *CO (**Figure R4-3b** and **e**), due to antibonding feature of $d_{xz}/d_{yz}-2\pi^*$ bond, which will effectively promote the hydrogenation of CO reduction intermediates. Due to different orbital electron configuration between LS and HS- Co²⁺, the injected electrons from electrode would result in two different roles for CORR, the detailed descriptions could be found in the manuscript (line 300-304). “In the case of LS-Co²⁺-CO, the electrons from the electrode will fill in the anti-bonding orbital of $3d_z^2-5\sigma$ bond, resulting in weakened *CO adsorption strength, unfavorable for *CO reduction. While in the case of HS-Co²⁺-CO, the electrons from the electrode will fill in the bonding orbital of $3d_{xz}/d_{yz}-2\pi^*$ bond, weakening the C-O bond in *CO, which can effectively promote *CO hydrogenation.” Overall, the increase in catalytic performance comes from the change in orbital electronic configuration caused by the distortion of CoPc molecules. To be specific, compared to the case of LS-Co²⁺, CO weakly adsorbs on HS-Co²⁺ by the weak $3d_z^2-5\sigma$ bond, while more electrons can

transfer from Co to *CO via π back-donation (via $d_{xz}/d_{yz}-2\pi^*$ bond) for HS-Co²⁺, enabling more electron accumulation on the $2\pi^*$ orbital of *CO, weakening the C-O bond in *CO, which can effectively promote *CO hydrogenation. The *operando* ATR-SEIRAS results show that “the *CO vibrational frequency on M-CoPc-400 is slightly lower than that on B-CoPc-400 (1911 and 1951 cm⁻¹ at 0 V vs. RHE), indicating a slightly lower binding strength of B-CoPc-400 towards CO”, which is consistent with the DFT calculation that the adsorption of HS Co on CO is weakened. Moreover, the weakened C-O bond is beneficial for the subsequent hydrogenation process, which is the RDS of CORR to produce CH₃OH. Therefore, the experiments and the DFT calculations are not contradictory.

Figure R4-3. *Operando* ATR-SEIRAS and KIE measurements. *Operando* ATR-SEIRAS spectra of CO reduction over (a, c) B-CoPc-400 and (b, d) M-CoPc-400 in CO-saturated 0.5 M KOH. The spectra were collected at constant potentials with 0.1 V interval in the cathodic direction from OCP to -0.9 V (vs. RHE). The *operando* ATR-SEIRAS spectra recorded in Ar-saturated 0.5 M KOH are shown in Figure S13. (e) Potential dependent CO stretching peak intensity. (d) KIE of H/D in CORR to CH₃OH at -0.6 V (vs. RHE) over B-CoPc-400 and M-CoPc-400.

Figure R5. CORR mechanism. (A) Free energy diagram of CORR over LS-Co²⁺ (M-CoPc-400) and HS-Co²⁺ (B-CoPc-400). (B) The charge density distributions of LS-Co²⁺-CO and HS-Co²⁺-CO. Blue, yellow, pink and gray balls represent N, C, Co and O atoms, respectively, and the faint yellow and cyan regions refer to the increased and decreased charge density. Interactions between CO molecular frontier orbitals (5σ and 2π*) and the 3d orbital of (C) LS-Co²⁺ and (D) HS-Co²⁺ site. Dot in black of (C) and (D) is the electron from electrode under cathodic bias. (E) Schematic of σ and π donation bonds between CO and 3d orbital of LS-Co²⁺ and HS-Co²⁺. The size of the arrow indicates the binding strength of 3d_{z²}-5σ (in red color) and 3d_{xz}/d_{yz}-2π* (in green color), which also indicates the magnitude of the electron migration.

3. The expression in line 246 is incorrect: “suggesting that unpaired electron exists in M-CoPc-RT and M-CoPc-RT.”

Response: We thank the reviewer for pointing out this typo. The expression “suggesting that unpaired electron exists in M-CoPc-RT and M-CoPc-RT.” has been revised to “suggesting that unpaired electron exists in M-CoPc-RT and B-CoPc-RT.”

REVIEWERS' COMMENTS

Reviewer #1 (Remarks to the Author):

The authors have addressed most of my concerns and this manuscript is suggested to be published in Nature Communications.

One minor concern: the authors may misconstrue the questions in Q4. The reviewer never doubted the C-H assignment in the recommended references (J. Am. Chem. Soc. 2022, 144, 6613, Chem 2021, 7, 1297, Angew. Chem. Int. Ed. 2022, 61, e202206233). The reviewer's point is that those references do not show the C-H vibrations in *CHO species. The three peaks observed in the range of 2850~2950 cm⁻¹ originates from the different vibrational modes of C-H in -CH₃ and -CH₂ groups (J. Am. Chem. Soc. 2022, 144, 6613), rather than being associated with *CHO. The C-H in *CHO species is supposed to exhibit a sole stretching mode, however, the authors documented three different modes (2850~3000 cm⁻¹) in their IR results.

Reviewer #2 (Remarks to the Author):

The authors have cleared my concerns, thus, this reviewer recommend the publication of this manuscript.

Reviewer #4 (Remarks to the Author):

The reviewer appreciates the efforts made by the authors, most of the concerns have been addressed. The manuscript can be published.

Response to Reviewers

We are very grateful to the critical comments and constructive suggestions provided by the reviewers, which shall significantly help to improve the quality of our work. Our manuscript has been revised accordingly, and the changes are highlighted by yellow colour in the text. The following list our responses to the comments from the reviewers. All the original comments are given in blue colour (italic) and our responses in black colour.

Reviewer #1 (Remarks to the Author):

The authors have addressed most of my concerns and this manuscript is suggested to be published in Nature Communications.

One minor concern: the authors may misconstrue the questions in Q4. The reviewer never doubted the C-H assignment in the recommended references (J. Am. Chem. Soc. 2022, 144, 6613, Chem 2021, 7, 1297, Angew. Chem. Int. Ed. 2022, 61, e202206233).

*The reviewer's point is that those references do not show the C-H vibrations in *CHO species. The three peaks observed in the range of 2850~2950 cm⁻¹ originates from the different vibrational modes of C-H in -CH₃ and -CH₂ groups (J. Am. Chem. Soc. 2022, 144, 6613), rather than being associated with *CHO. The C-H in *CHO species is supposed to exhibit a sole stretching mode, however, the authors documented three different modes (2850~3000 cm⁻¹) in their IR results.*

Response: We sincerely appreciate the reviewer for the time and efforts spent in evaluating our manuscript. We are sorry for the unclear assignment of C-H vibrations in the range of 2850~2950 cm⁻¹. The description has been revised as “It is worth to note that the C-H bands between 3000 cm⁻¹ and 2800 cm⁻¹ originating from different vibrational modes in -CH₃ and CH₂ over B-CoPc-400 are inverted peaks, different from the positive peaks over M-CoPc-400.”

Reviewer #2 (Remarks to the Author):

The authors have cleared my concerns, thus, this reviewer recommend the publication of this manuscript.

Response: We sincerely appreciate the reviewer for the time and efforts spent in evaluating our manuscript.

Reviewer #4 (Remarks to the Author):

The reviewer appreciates the efforts made by the authors, most of the concerns have been addressed. The manuscript can be published.

Response: We sincerely appreciate the reviewer for the time and efforts spent in evaluating our manuscript.